# GPCR kinase knockout cells reveal the impact of individual GRKs on arrestin binding and GPCR regulation

J. Drube [1,4], R. S. Haider [1,4], E. S. F. Matthees [1], M. Reichel [1], J. Zeiner[2], S. Fritzwanker[3], C. Ziegler[1], S. Barz[1], L. Klement[1], J. Filor[1], V. Weitzel[1], A. Kliewer[3], E. Miess-Tanneberg[3], E. Kostenis [2], S. Schulz [3] & C. Hoffmann [1✉]

G protein-coupled receptors (GPCRs) activate G proteins and undergo a complex regulation by interaction with GPCR kinases (GRKs) and the formation of receptor–arrestin complexes. However, the impact of individual GRKs on arrestin binding is not clear. We report the creation of eleven combinatorial HEK293 knockout cell clones lacking GRK2/3/5/6, including single, double, triple and the quadruple GRK knockout. Analysis of β-arrestin1/2 interactions for twelve GPCRs in our GRK knockout cells enables the differentiation of two main receptor subsets: GRK2/3-regulated and GRK2/3/5/6-regulated receptors. Furthermore, we identify GPCRs that interact with β-arrestins via the overexpression of specific GRKs even in the absence of agonists. Finally, using GRK knockout cells, PKC inhibitors and β-arrestin mutants, we present evidence for differential receptor–β-arrestin1/2 complex configurations mediated by selective engagement of kinases. We anticipate our GRK knockout platform to facilitate the elucidation of previously unappreciated details of GRK-specific GPCR regulation and β-arrestin complex formation.

[1] Institut für Molekulare Zellbiologie, CMB – Center for Molecular Biomedicine, Universitätsklinikum Jena, Friedrich-Schiller-Universität Jena, Hans-Knöll-Straße 2, D-07745 Jena, Germany. [2] Molecular, Cellular and Pharmacobiology Section, Institute for Pharmaceutical Biology, University of Bonn, Nussallee 6, 53115 Bonn, Germany. [3] Institut für Pharmakologie und Toxikologie, Universitätsklinikum Jena, Friedrich-Schiller-Universität Jena, Drackendorfer Straße 1, D-07747 Jena, Germany. [4] These authors contributed equally: J. Drube, R. S. Haider. ✉email: carsten.hoffmann@med.uni-jena.de

G protein-coupled receptors (GPCRs) constitute the largest family of membrane receptors in human physiology comprising more than 800 identified members. GPCRs regulate multitudes of physio- and pathophysiological processes and are well-established targets for pharmacological intervention. A recent review listed 134 GPCRs and about 50 additional GPCR-signalling related proteins which are directly targeted by Food and Drug Administration-approved drugs[1].

The diverse stimuli recognised by GPCRs induce conformational changes within the receptor, which activate distinct signalling pathways[2]. As opposed to the large number of GPCRs, the intracellular signalling molecules are less diverse. Besides G proteins, GPCR kinases (GRKs) and arrestins are the most immediate GPCR-interacting molecules[3,4]. The human genome contains seven genes that encode for different GRK isoforms: GRK1–7. The two visual GRKs (GRK1 and 7) are specifically expressed in the retina to facilitate the shutoff of photopigment signalling. Similarly, the expression of GRK4 predominantly occurs in specific tissues (e.g. testis or the heart)[5]. Of the four arrestin genes that are conserved in vertebrates (arrestin-1 or visual arrestin, arrestin-2 or β-arrestin1, arrestin-3 or β-arrestin2 and arrestin-4 or cone arrestin), two isoforms, namely arrestin-1 and -4, aid in photopigment desensitisation and are restrictively expressed in rod and cone cells. Hence, the regulation of hundreds of non-visual GPCRs is hypothesised to be orchestrated by just six ubiquitously expressed proteins: four GRKs (GRK2, 3, 5 and 6) and two arrestin isoforms (β-arrestin1 and 2)[6].

To explain this apparent imbalance, the phosphorylation barcode hypothesis for receptor–arrestin interactions was developed[7–9]. Since GRK-induced GPCR phosphorylation is the basis for high-affinity β-arrestin binding, we anticipate that individual GRK isoforms shape the GPCR signalling response in a cell- and tissue-specific manner.

The relative selectivity of ligands to favour a certain pathway at the expense of others was termed functional selectivity or biased agonism[6,10,11]. The recognition that either G protein- or arrestin-supported pathways[12] can contribute to pathophysiological conditions or drug-associated side effects[10,13] triggered an intense search for biased ligands. GRKs act as essential mediators and define β-arrestin functions via ligand-specific GPCR phosphorylation or preferential coupling to certain active receptor states. Structural biology greatly contributed to our understanding of receptor conformational changes, which lead to the interaction with either G proteins or arrestins. For arrestins, a multi-step GPCR binding model[14] was proposed back in 1993. This mechanism involves the recognition of receptor phosphorylation[15] and the engagement of the arrestin finger loop region (FLR)[16,17]. However, little is currently known about the impact of individual GRKs on arrestin binding.

The ubiquitous expression of GRK2, 3, 5 and 6 obscures the elucidation of the roles of individual GRKs in receptor phosphorylation. Until now, siRNA/shRNA[18–20] or CRISPR/Cas9 approaches targeting only a certain subset of relevant GRKs[21], and the utilisation of GRK inhibitors were the only strategies used to study their impact on living cell function. Yet, in combination with phosphosite-specific antibodies[22,23] or mass spectrometry[24], contributions of individual GRKs to the phosphorylation of certain receptors were elucidated to some degree[8].

Nevertheless, the remaining expression of the targeted GRK(s) in knockdown approaches, or potential off-target effects of pharmacological intervention preclude the unambiguous interpretation of obtained results. Thus, a comprehensive elucidation of single GRK contributions to the arrestin-dependent regulation of GPCR signalling, internalisation and trafficking remains elusive.

In this study, we present a cellular platform to investigate the individual roles of GRK2, 3, 5 and 6 in these processes. We have created a panel of eleven combinatorial HEK293 GRK knockout clones, which enable us to analyse the GRK contributions to GPCR phosphorylation, recruitment of β-arrestin1 and 2, as well as receptor internalisation in unprecedented detail.

## Results

**GRK knockout cells: a viable cellular platform to assess individual GRK contributions.** Utilising the CRISPR/Cas9 technology, we engineered HEK293 single-cell clones with knockouts (KO) of GRKs. We created single KOs of GRK2 (ΔGRK2), GRK3 (ΔGRK3), GRK5 (ΔGRK5), GRK6 (ΔGRK6), two double KOs ΔGRK2/3 and ΔGRK5/6 and a quadruple KO of GRK2, 3, 5 and 6 (ΔQ-GRK). Additionally, we established four triple KO cell clones (ΔGRK3/5/6, ΔGRK2/5/6, ΔGRK2/3/6 and ΔGRK2/3/5) with the endogenous expression of one remaining GRK. To compare experiments using these KOs, we furthermore subjected HEK293 cells to the CRISPR/Cas9 process without the addition of any gRNAs (Control).

The KOs were confirmed by Western blot analysis (Fig. 1a) and further validated by the functional studies contained in this manuscript. Morphology as revealed by phase-contrast microscopy of cultured cells and cell growth (Supplementary Fig. 1a–c) were only mildly affected in some of the clones. Expression levels of the untargeted GRKs in the obtained cell clones remained virtually unchanged compared to Control (Supplementary Fig. 1d). Notably, we did not assess the expression levels of other kinases that might influence GPCR regulation, thus we cannot exclude that the presented cell lines feature the expression of e.g. GRK1, 4 and/or 7.

In order to test the effect of our ΔGRK-clones and to investigate potentially altered kinase activity of GRKs untargeted in a specific KO cell line, we revisited and analysed agonist promoted μ-opioid receptor (MOP) phosphorylation. This receptor system was deliberately chosen, as GRK contributions were already extensively studied[21,25] and the abundance of available phosphosite-specific antibodies allowed for complete elucidation of [D-Ala2, N-MePhe4, Gly-ol]-enkephalin (DAMGO)-induced receptor phosphorylation (Supplementary Fig. 2a, b). We successfully identified $T^{376}$ as a specific target of GRK2 and 3. In line with our previous findings[25], $T^{370}$, $S^{375}$ and $T^{379}$ seem to be phosphorylated by all four GRKs albeit to different extents. In ΔQ-GRK, the phosphorylation of these sites was completely abolished, confirming their role as GRK target sites. In contrast, $S^{363}$, a known PKC phosphorylation site[25,26] retained its strong phosphorylation signal in ΔQ-GRK cells (Supplementary Fig. 2a, b). This analysis confirms the general functionality of our clones for phosphorylation studies and underlines the unaltered activity of kinases not targeted in our KO approach.

Further, we investigated MOP internalisation in Control, ΔGRK2/3, ΔGRK5/6 and ΔQ-GRK stably expressing the receptor by confocal microscopy and surface enzyme-linked immunosorbent assay (ELISA) (Supplementary Fig. 2c, d). After stimulation, MOP was internalised in Control and ΔGRK5/6, but remained at the cell surface in ΔGRK2/3 and ΔQ-GRK. Our findings confirm that all four analysed GRKs are able to act on the MOP, but exclusively GRK2 and 3 are able to phosphorylate $T^{376}$ and further drive the internalisation of the receptor.

**GRK2, 3, 5 and 6 are able to individually induce the formation of b2AR–β-arrestin complexes.** As the availability of tools for the analysis of site-specific receptor phosphorylation is limited across the GPCR superfamily, we utilised the universal GPCR adaptor proteins β-arrestin1 and 2 to analyse the contributions of individual GRKs to receptor regulation. The schematic in Fig. 1b

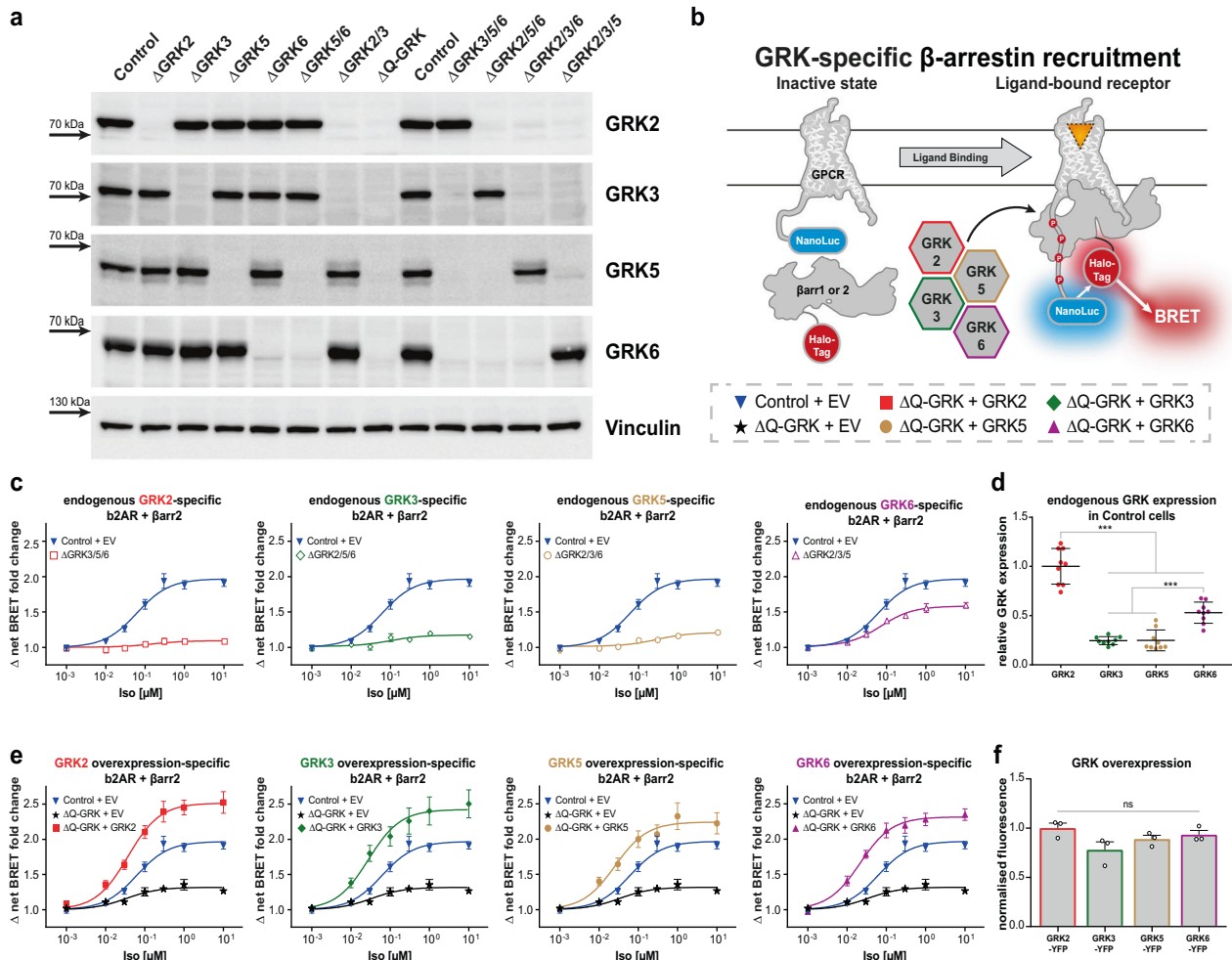

**Fig. 1 GRK knockout cells enable the characterisation of GRK-specific β-arrestin recruitment. a** Single (ΔGRK2, 3, 5 and 6), double (ΔGRK2/3 or 5/6), triple (ΔGRK3/5/6, 2/5/6, 2/3/6 and 2/3/5) and quadruple (ΔQ-GRK) GRK knockout cells were generated using the CRISPR/Cas9 technology and established as single-cell clones. The absence of GRK2, 3, 5 or 6 was confirmed by Western blot analysis. A representative blot of $n = 4$ independent experiments is shown (for quantification, see Supplementary Fig. 1d). **b** Schematic depiction of the performed NanoBRET β-arrestin (βarr) recruitment assay and colour-coding for GRK-specific conditions used throughout the paper. The Halo-Tag-βarr fusion protein is recruited to a NanoLuciferase (NanoLuc)-tagged GPCR upon agonist activation and subsequent receptor phosphorylation. The resulting change in proximity of the Halo-Tag and the NanoLuc increases measured BRET ratios, enabling the agonist concentration-dependent analysis of βarr recruitment. **c** Halo-Tag-βarr2 recruitment to the b2AR-NanoLuc upon stimulation with isoprenaline (Iso) in cells expressing all endogenous GRKs (Control + empty vector (EV)) or only one remaining endogenous GRK (triple knockout cells ΔGRK3/5/6, ΔGRK2/5/6, ΔGRK2/3/6 and ΔGRK2/3/5). **d** The relative GRK protein expression in Control cells determined by Western blot of $n = 3$ independent lysates as described in Reichel et al.[27]. Data are depicted as mean ± SEM of $n = 3$ independent blots. GRK expression levels were compared using ANOVA and two-sided Tukey's test (*$p < 0.05$; **$p < 0.01$; ***$p < 0.001$). **e** βarr2 recruitment to the b2AR in quadruple GRK knockout cells (ΔQ-GRK), overexpressing a single GRK (ΔQ-GRK + GRK) or expressing all endogenous GRKs (Control + EV). BRET data in (**c**) and (**e**) are presented as Δ net BRET fold change, mean of $n = 3$ independent experiments ± SEM. For better comparison, the Control and ΔQ-GRK curves are shown multiple times. **f** GRK–YFP fusion proteins were transfected in ΔQ-GRK and YFP fluorescence was measured to confirm similar expression levels of all transfected GRKs. YFP fluorescence was compared using ANOVA and two-sided Tukey's test (ns not significant). Corresponding experiments, confirming the catalytic activity of GRK–YFP fusion constructs are shown in Supplementary Fig. 3a. Measured fluorescence is depicted as a mean of $n = 3$ independent experiments + SEM as normalised fluorescence. All exact $p$ values, test statistics, effect sizes, confidence intervals and degrees of freedom are provided in the Source Data files.

depicts the established bioluminescence resonance energy transfer (BRET)-based in cellulo β-arrestin recruitment assay, allowing us to reveal functional, GRK-specific GPCR phosphorylation. In this BRET assay, a NanoLuciferase (NanoLuc) is fused to the receptor C-terminus and serves as the energy donor in order to observe the association of β-arrestin constructs tagged with a Halo-Tag and labelled with the Halo 618 ligand.

First, we studied the GRK-specific interactions between the β2 adrenergic receptor (b2AR) and β-arrestin2 utilising the endogenous expression of GRKs in various ΔGRK cells. At

endogenous expression levels of all four GRKs (Control), β-arrestin2 showed clear isoprenaline (Iso)-induced recruitment to the b2AR (Fig. 1c). In comparison, β-arrestin recruitment was substantially reduced when recorded in triple GRK KO cell lines, only featuring the endogenous expression of one individual GRK (Fig. 1c, ΔGRK3/5/6, ΔGRK2/5/6, ΔGRK2/3/6 and ΔGRK2/3/5). While endogenous expression of either GRK2, 3 or 5 induced only minimal BRET changes, these were, nevertheless, sufficient to detect a ligand-dependent increase in β-arrestin recruitment. To evaluate whether recorded BRET data actually describes a

molecular GPCR–β-arrestin interaction, all recruitment data in this study have been subjected to statistical analysis (Supplementary Table 1). Hence, only those data sets that show a significant increase for the condition stimulated with the highest ligand concentration compared to vehicle addition, are further interpreted as functional β-arrestin recruitment. Additionally, concentration–response curves were only fitted for these conditions. The highest amount of β-arrestin2 recruitment to the b2AR was found in ΔGRK2/3/5 cells, specifically induced by the endogenous expression of GRK6.

These data have to be evaluated in the context of endogenous GRK expression levels in Control cells. Hence, we conducted Western blot analysis to assess the relative endogenous expression levels of GRK2, 3, 5 and 6 as elaborated in Reichel et al.[27] and found that the cytosolic GRK2 and the membrane-associated GRK6 are the most abundant (Fig. 1d). Interestingly, we identified GRK6 as the main mediator of Iso-promoted β-arrestin2 recruitment to the b2AR (Fig. 1c), when measured under the endogenous expression of GRKs. This suggests that specific GRK isoforms exhibit different affinities for coupling to the same GPCR and require differential expression levels to facilitate functional receptor regulation.

Since these findings specifically reflected on the affinities and endogenous expression levels of GRKs, we analysed the molecular capability of each individual GRK to induce b2AR–β-arrestin2 complex formation via re-introduction into ΔQ-GRK (Fig. 1e). The relative expression of transfected GRKs was assessed fluorometrically (Fig. 1f). Via the introduction of a C-terminal YFP fusion into the identical vector backbone and subsequent equimolar transfection of GRK–YFP constructs, we confirmed similar expression levels of the transfected kinases. To allow for this comparison, the GRK–YFP fusion proteins were characterised with at least the same capability to mediate GPCR–β-arrestin interactions as their untagged counterparts, used in all other experiments (Supplementary Fig. 3a). Additionally, the relative degree of overexpression was quantified via Western blot analysis (Supplementary Fig. 3b, c). Using this controlled overexpression of individual GRKs in ΔQ-GRK, all four kinases showed a similar effect on b2AR regulation: each individual GRK isoform enhanced the b2AR–β-arrestin recruitment to higher levels than induced by the combined endogenous expression of GRKs in Control cells (Fig. 1e). Interestingly, we still encountered measurable β-arrestin2 recruitment in the absence of GRKs (ΔQ-GRK + EV). This could be explained by the inherent affinity of β-arrestin2 towards ligand-activated, yet unphosphorylated GPCRs.

These findings clarify that all four tested GRKs are able to individually mediate high-affinity β-arrestin2 binding to the b2AR and that their relative tissue expression ultimately defines their specific contributions to this process.

Since all GRKs have been shown to induce similar levels of β-arrestin recruitment, we investigated whether isoform-specific phosphorylation of the b2AR might still have a pronounced effect on the conformational changes that occur during arrestin activation. To address this, we overexpressed each individual GRK isoform alongside the untagged b2AR and an intramolecular β-arrestin2-FlAsH5-NanoLuc BRET biosensor. We found comparable β-arrestin2 conformational changes for all GRKs (Fig. 2a). Note, that an equivalent FRET sensor was published previously[28] and that more details on the intramolecular BRET sensor will be published elsewhere.

We further utilised the b2AR, as a model receptor regulated by all four tested GRK isoforms, to test the effect of endogenous ligands and pharmacological inhibition on GRK-specific β-arrestin-coupling processes. The application of the endogenous ligands epinephrine and norepinephrine resulted in overall lower GRK-specific β-arrestin2 recruitment to the b2AR as compared to

Iso (Supplementary Fig. 4a, b compare with Fig. 1e). Although the relative efficacies of individual GRKs to mediate epinephrine- and norepinephrine-induced β-arrestin2 binding was unchanged in comparison to Iso (ΔQ-GRK + GRK > Control + EV > ΔQ-GRK + EV), we observed a left shift of the measured concentration-response curves specifically for GRK6 (Fig. 2b–d). This significant increase in potency to elicit β-arrestin2 recruitment was observed for both endogenous ligands, although it was shown that epinephrine acts as a full agonist, whereas norepinephrine only partially activates the b2AR[29]. This might have implications for the tissue-specific regulation of b2AR, as lower ligand concentrations might be sufficient to desensitise the receptor in tissues with relatively higher GRK6 expression.

Since we were able to measure GRK-specific β-arrestin recruitment, we hypothesise that this assay is also suitable to characterise the specificity of GRK inhibitors in a cellular system. Indeed, we were able to record the concentration-dependent inhibition of β-arrestin2 recruitment to the receptor by cmpd101 (a known GRK2 family inhibitor) only in cells expressing GRK2 or 3 (Fig. 2e). This demonstrates cmpd101 selectivity by the lack of inhibition in cells overexpressing GRK5 or 6. When performing the analogous experiment using pindolol as a potent antagonist of the b2AR, we recorded an inhibition of β-arrestin2 recruitment regardless of GRK (over-) expression (Fig. 2f). Thus, we present a cell-based GRK-inhibitor screening platform utilising ΔQ-GRK.

### ΔQ-GRK cells reveal GRK-specificity of β-arrestin1 and 2 recruitment to different GPCRs.
To investigate the GRK-specificity of GPCR regulation, we compared the individual molecular capabilities of GRK2, 3, 5 and 6 to facilitate β-arrestin recruitment across 12 different GPCRs: angiotensin II type 1 receptor (AT1R), b2AR, b2AR with an exchanged C-terminus of the vasopressin 2 receptor (b2V2), complement 5a receptor 1 (C5aR1), muscarinic acetylcholine receptors (M1R, M2R, M3R, M4R and M5R), MOP, parathyroid hormone 1 receptor (PTH1R) and vasopressin 2 receptor (V2R). This receptor panel was deliberately selected to feature receptors with divergent lengths of intracellular loop 3 (IL3, ranging from 3 to 213 amino acids) and C-termini (ranging from 8 to 105 amino acids), predominant coupling to different G proteins (Supplementary Table 2), as well as GPCRs that have been shown to be additionally regulated by second messenger kinases.

As representative examples of our findings, the GRK-selective β-arrestin1 and 2 recruitment to the M5R and PTH1R are depicted in Fig. 3a, b and Fig. 3c, d, respectively (data for all receptors are shown in Supplementary Figs. 5 and 6). Both receptors were able to induce robust, agonist-dependent β-arrestin1 and 2 recruitment in Control cells, which was significantly reduced in ΔQ-GRK. In the case of the M5R, β-arrestin1 recruitment was completely abolished in ΔQ-GRK. Still, a major difference in GRK-selectivity of the two receptors was found using this approach: the individual overexpression of GRK2, 3, 5 and 6 significantly increased β-arrestin recruitment to the PTH1R in ΔQ-GRK, whereas GRK5 and 6 were unable to facilitate M5R–β-arrestin complex formation. These findings were additionally confirmed in triple GRK KO cell lines and the ΔGRK2/3 and ΔGRK5/6 family KOs (Supplementary Fig. 7a, b). Interestingly, the endogenous expression of GRK2 and 3 in ΔGRK3/5/6 and ΔGRK2/5/6 was sufficient to increase the measured β-arrestin2 recruitment in comparison to ΔQ-GRK for both receptors. This finding essentially confirms the functionality of these two triple GRK KO cell lines and suggests that the M5R and PTH1R require lower amounts of GRK2 or 3 in order to be efficiently regulated in comparison to the b2AR

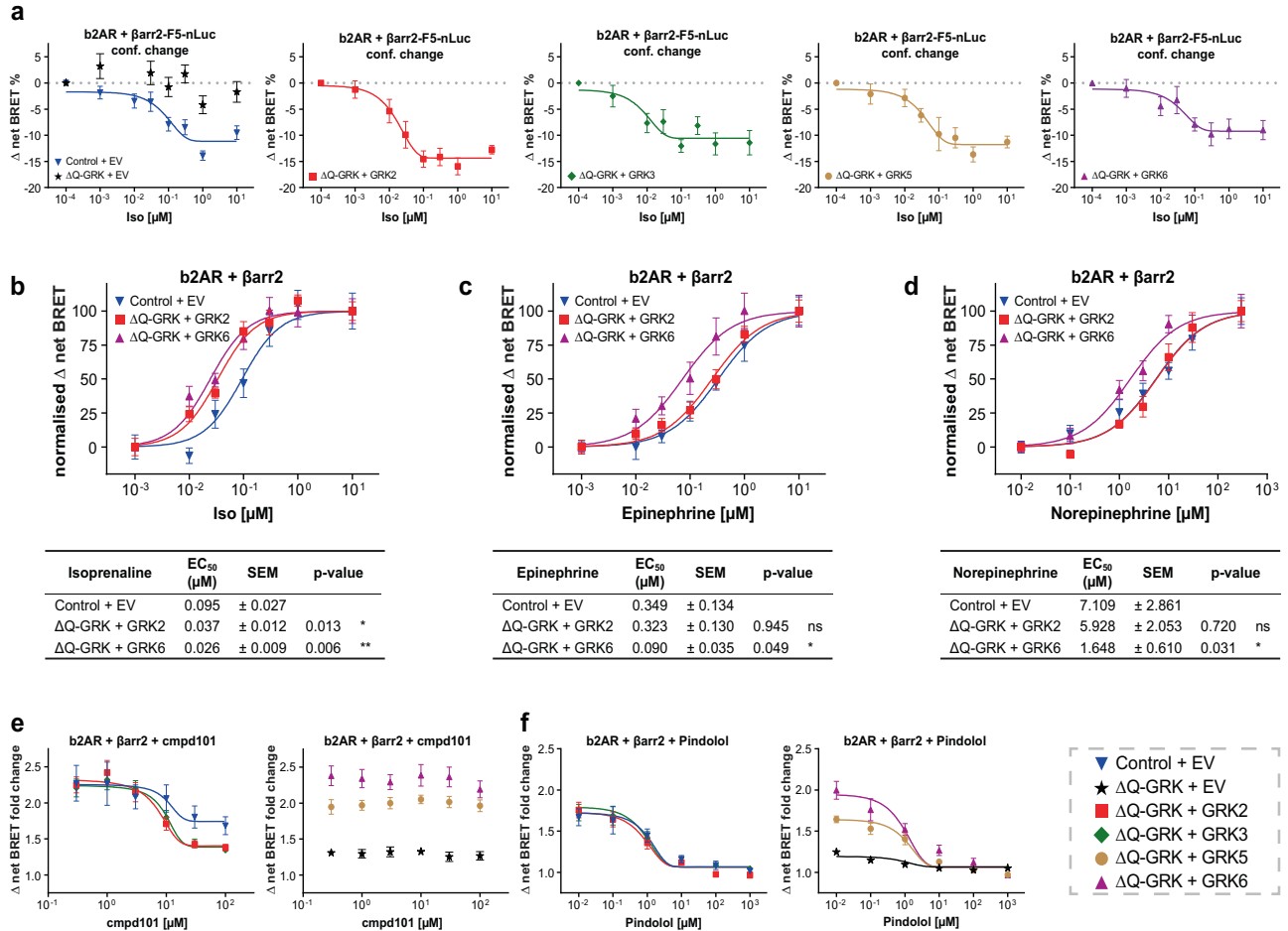

**Fig. 2 GRK knockout cells are a viable cellular platform to assess various features of b2AR–β-arrestin complex formation. a** Analysis of β-arrestin2 (βarr2) conformational changes. ΔQ-GRK or Control cells were transfected with an untagged b2AR expression construct and the βarr2-F5-NanoLuc conformational change biosensor (more details on the intramolecular BRET sensor will be published elsewhere), in the absence or presence of GRKs as noted and stimulated with isoprenaline (Iso). Conformational change data are shown as Δ net BRET change in per cent, mean of $n = 3$ independent repetitions ± SEM. **b–d** The recruitment of βarr2 to the b2AR following stimulation with Iso (**b**), Epinephrine (**c**) or Norepinephrine (**d**) in presence of all endogenous GRKs or individually overexpressed GRK2 or GRK6 in ΔQ-GRK as indicated. Data are depicted as a mean of $n = 3$ independent experiments ± SEM and normalised to individual maxima. The tables below depict the $EC_{50}$ ± SEM of the corresponding concentration-response curves. The $EC_{50}$ of the indicated conditions were compared to the $EC_{50}$ in Control using ANOVA and two-sided Dunnett's test (*$p < 0.05$; **$p < 0.01$; ns not significant). **e, f** Utilisation of the βarr recruitment assay for specificity determination of the GRK inhibitor cmpd101 in living cells. ΔQ-GRK or Control cells were transfected with b2AR-NanoLuc, Halo-Tag-βarr2 and either GRK2, 3, 5, 6 or EV as noted. The cells were incubated with different concentrations of cmpd101 (**e**) or the b2AR antagonist pindolol (**f**) for 10 min prior to stimulation with 1 µM Iso. The recruitment-induced BRET changes were measured and calculated as Δ net BRET change in per cent, represented as the mean of $n = 4$ independent experiments ± SEM. All exact $p$ values, test statistics, effect sizes, confidence intervals and degrees of freedom are provided in the Source Data files.

(Fig. 1c). As indicated by the experiments shown in Fig. 3b, ΔGRK cell lines only featuring the expression of GRK5 and/or 6 did not increase the β-arrestin2 recruitment to the M5R as in comparison to ΔQ-GRK.

Further, we employed confocal live-cell microscopy to assess the dependency of PTH1R and M5R internalisation on endogenous GRK levels in Control, ΔGRK2/3 and ΔGRK5/6 as well as in ΔQ-GRK. Under basal conditions, β-arrestin2 is located in the cytosol, M5R and PTH1R in the cell membrane and Rab5 (early endosome marker) in endosomes (Fig. 3e, f basal). As expected, the M5R was not able to induce β-arrestin2 translocation in the absence of GRK2 and 3 (ΔGRK2/3 and ΔQ-GRK) (Fig. 3e).

The quantification of co-localisation between the M5R and β-arrestin2 (Fig. 3g) confirms our findings of Fig. 3b and Supplementary Fig. 7a. Analysis of M5R co-localisation with Rab5 (as a surrogate measurement for receptor internalisation

and initial trafficking) reveals that this interaction translates to functional receptor internalisation only in the presence of GRK2 and 3 (Fig. 3e, g). For the PTH1R, we were able to detect ligand-induced co-localisation with β-arrestin2 or Rab5 in all conditions expressing GRKs (Fig. 3f, h). Interestingly, the agonist-stimulated PTH1R was still able to induce a slight membrane translocation of β-arrestin2 in ΔQ-GRK, confirming the GRK-independent affinity of β-arrestin2 toward the ligand-activated receptor (Fig. 3d). The results obtained for the PTH1R using endogenous GRK expression were verified in a reciprocal experiment overexpressing single GRKs in ΔQ-GRK for β-arrestin1 and 2 (Supplementary Fig. 7c–f).

These apparent differences in GRK-specific β-arrestin recruitment, as exemplified by the M5R and the PTH1R, were encountered multiple times during our analysis across twelve different GPCRs (Supplementary Fig. 5). Via statistical multiple comparisons of BRET fold changes at saturating ligand concentrations for each of the tested

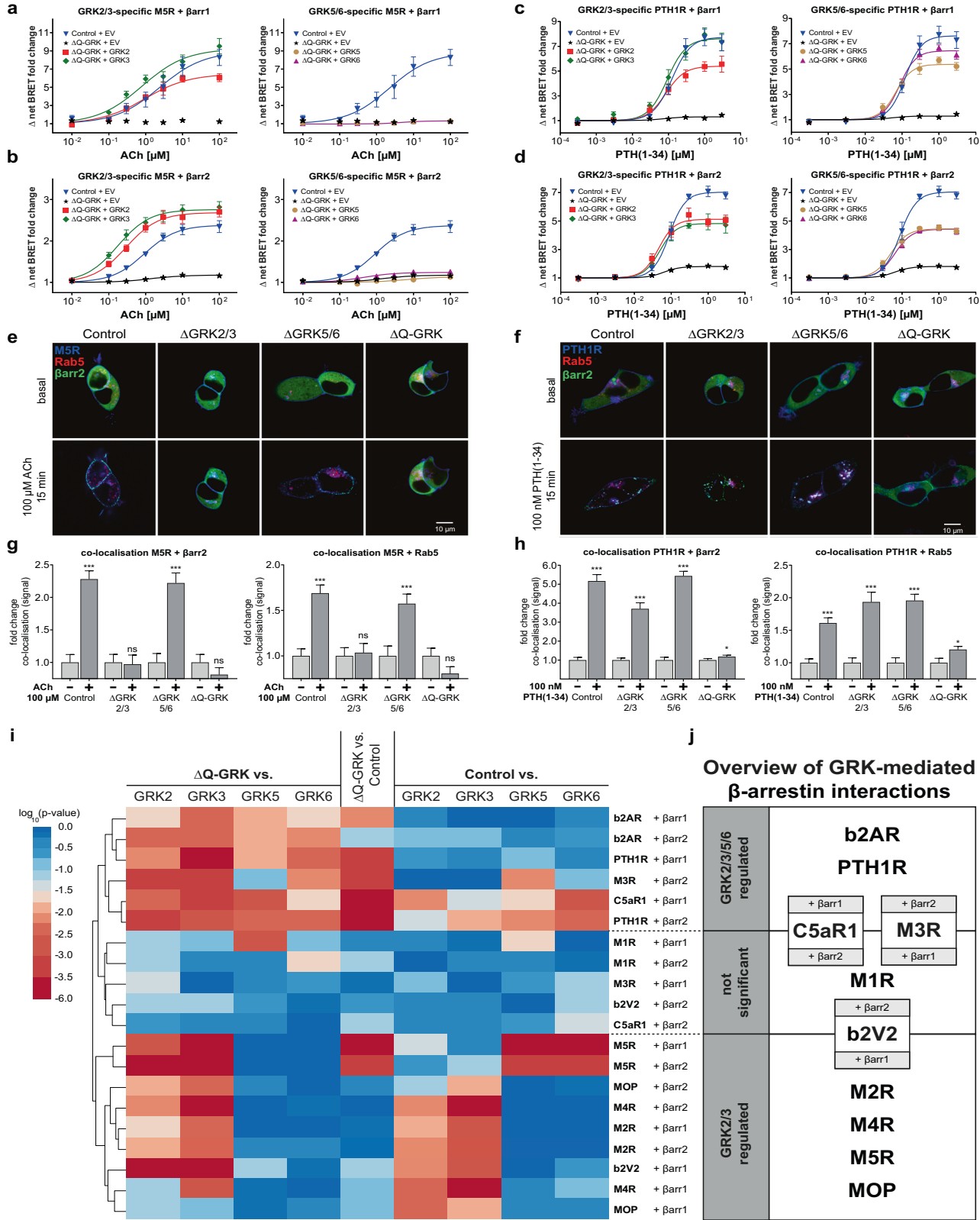

conditions (Control + EV, ΔQ-GRK + EV, ΔQ-GRK + GRK2, ΔQ-GRK + GRK3, ΔQ-GRK + GRK5 and ΔQ-GRK + GRK6), we were able to cluster the respective GPCR–β-arrestin pairs into groups, depending on the found GRK-selectivity (Fig. 3i, j; Supplementary Table 3). Here, we identified several subsets of GPCRs: receptors for which β-arrestin interaction is mediated by overexpression of (i) any

GRK (b2AR, PTH1R, C5aR1 + β-arrestin1 and M3R + β-arrestin2) or (ii) GRK2 or 3 only (M2R, M4R, M5R, MOP and b2V2 + β-arrestin1). Within our tested GPCRs, we did not observe β-arrestin interaction mediated exclusively by GRK5 or 6. A third group is comprised of receptor–β-arrestin pairs, which did not consistently show significant differences between the tested conditions and hence

**Fig. 3 ΔQ-GRK cells reveal GRK-specificity of β-arrestin1 and 2 recruitment to different GPCRs and allow assessment of GRK-dependent GPCR internalisation and β-arrestin2 translocation. a–d** GRK-specific β-arrestin (βarr)1 (**a**, **c**) or βarr2 (**b**, **d**) recruitment to the M5R upon acetylcholine (ACh) stimulation (**a**, **b**) or the PTH1R upon parathyroid hormone 1–34 (PTH(1-34)) stimulation (**c**, **d**). Shown are concentration-response curves depicted as Δ net BRET fold change, mean of n = 3 independent experiments ± SEM. The panels display the recruitment in presence of either GRK2 or 3 or GRK5 or 6. For better comparison, the Control and ΔQ-GRK curves are shown multiple times. **e**, **f** Control, ΔGRK2/3, ΔGRK5/6 and ΔQ-GRK cells were transfected with either M5R-CFP or PTH1R-CFP (blue), the early endosome marker Rab5-mCherry (red) and βarr2-YFP (green) expression constructs. The cells were grown on coverslips and subjected to confocal live-cell microscopy. Shown are representative images, taken before and after 15 min of stimulation with either 100 μM ACh or 100 nM PTH(1–34), respectively. The normalised co-localisation of M5R (**g**) or PTH1R (**h**) with βarr2 or Rab5 was quantified using Squassh and SquasshAnalyst (number of images per respective condition; Control: M5R (39), PTH1R (38); ΔGRK2/3: M5R (35), PTH1R (32); ΔGRK5/6: M5R (36), PTH1R (33); ΔQ-GRK: M5R (38), PTH1R (55)). Data are presented as mean fold change in co-localisation signal + SEM. Statistical analysis was performed using a two-way mixed model ANOVA followed by a two-sided paired t-test (*p < 0.05; **p < 0.01; ***p < 0.001; ns not significant). **i** Clustering heatmap representing the statistical multiple comparisons of βarr recruitment data for ten different GPCRs. Conditions with overexpressed GRKs were tested against ΔQ-GRK + empty vector (EV) or Control + EV, as indicated. Additionally, ΔQ-GRK + EV as compared to Control + EV. BRET fold changes at saturating ligand concentrations of at least n = 3 independent experiments were compared using ANOVA and two-sided Bonferroni's test (Supplementary Table 3, data derived from Supplementary Fig. 5). Transformed unadjusted p values are plotted. GPCR–βarr pairs are clustered according to Canberra distance. **j** Overview of clustering from **i** in GPCR–βarr pairs regulated by any tested GRK (GRK2/3/5/6 regulated), by GRK2 or 3 only (GRK2/3 regulated) and a third group, which is comprised of GPCR–βarr pairs that do not consistently show significant differences between the tested conditions. All exact p values, test statistics, effect sizes, confidence intervals and degrees of freedom are provided in the Source Data files.

could not be definitively assigned to one of the first two groups (M1R, C5aR1 + β-arrestin2, M3R + β-arrestin1 and b2V2 + β-arrestin2). In case of C5aR1 + β-arrestin2 (Supplementary Fig. 5d) this behaviour is explained by exceptionally high β-arrestin2 recruitment in the absence of GRKs.

Interestingly, two distinct GPCRs, namely the AT1R and V2R, evaded the statistical grouping process. Both receptors exhibited apparently diminished, agonist-dependent β-arrestin recruitment in the presence of certain overexpressed GRKs as compared to their effects in ΔQ-GRK (Supplementary Fig. 5g, m). This finding was highly unexpected, therefore we further focussed on the elucidation of these GRK-dependent processes.

**GRK2, 3, 5 or 6 individually enable ligand-independent β-arrestin1 and 2 interactions with the V2R.** Besides previous intensive studies[30,31], the V2R and AT1R stood out unique in our in-depth, GRK subtype-specific β-arrestin recruitment assay (Fig. 4, Supplementary Fig. 5m). In the presence of all four endogenously expressed GRKs (Control), agonist stimulation induced clear recruitment of β-arrestin1 and 2 to the V2R (Fig. 4a, b; Supplementary Fig. 8a, b). In their absence (ΔQ-GRK) the recruitment of β-arrestins was reduced, as expected. Surprisingly, individual overexpression of GRK2, 3, 5 or 6 did not further increase the concentration-dependent, dynamic BRET change of the interaction. When comparing the respective BRET ratios measured before and after stimulation at ligand saturation (Fig. 4a, b; Supplementary Fig. 8a, b), we found that already the basal BRET ratios were remarkably increased in presence of overexpressed GRKs. This was unlike any other receptor investigated in Fig. 3i (Supplementary Figs. 5 and 6).

The concentration–response curves shown to this point reflect the fold change between the measured baseline and stimulated BRET ratios. The elevated baselines explain the unexpectedly low dynamic BRET changes, even though the absolute values of stimulated BRET ratios show a clear increase in the presence of overexpressed GRKs. Thus, we conclude that already the basal molecular interaction between the V2R and β-arrestins is increased under those conditions.

Two major interaction interfaces between GPCRs and β-arrestins have been proposed. Namely, the interaction mediated by phosphorylated intracellular domains of the receptor (e.g. C-terminus and IL3) only ("hanging" complex)[17,32], as well as the additional insertion of the arrestin FLR into the intracellular cavity of the GPCR ("core" complex)[16]. Since we expected the unstimulated V2R to be in an inactive conformation, we hypothesised that the interaction between β-arrestins and the intracellular cavity of the GPCR is prevented and therefore occurs in a "hanging" conformation. If this hypothesis was correct, deletion of the β-arrestin FLR should not impair the measured association with V2R. Thus, we analysed β-arrestin recruitment with biosensors lacking the FLR (β-arrestin1/2-dFLR) (Fig. 4c, d; Supplementary Fig. 8c, d).

Indeed, for β-arrestin1-dFLR the baseline BRET measurements remained elevated, while agonist stimulation was not able to further increase the interaction between β-arrestin1-dFLR and the V2R (Fig. 4c, d). Thus, we propose ligand-independent pre-coupling of β-arrestin1 to the V2R in a "hanging" complex in presence of overexpressed GRKs. The remaining ligand-dependent increase in β-arrestin1 recruitment might be explained by ligand-activation of the pre-coupled "hanging" complex and subsequent engagement of the FLR to form a tight "core" complex (Fig. 4g). In contrast, β-arrestin2 pre-coupling was found to depend on the FLR (Supplementary Fig. 8c, d). These findings show that β-arrestin1 interacts with the V2R using an association with the C-terminus whereas β-arrestin2 seems to require both interactions, including the association of the FLR with the transmembrane helix bundle.

However, the kinase-dependent pre-coupling of β-arrestins to the receptor was not observed for the chimeric b2V2 (Supplementary Fig. 6). Hence, we conclude that the C-terminus is not solely responsible for the mediation of this effect.

Via confocal microscopy, we observed that the individual overexpression of GRK2 and 6 significantly increased the ligand-independent co-localisation between the V2R and Rab5 or β-arrestin1 in comparison to ΔQ-GRK (Fig. 4e, f; Supplementary Fig. 8e–h). This confirms that ligand-independent V2R–β-arrestin interactions, as facilitated by overexpressed GRKs, lead to functional receptor internalisation in line with the observations of Snyder et al.[33]. Furthermore, experiments conducted with GRK2 and 6 kinase-dead (KD; K220R, K215R, respectively) mutants (Supplementary Fig. 9) support that pre-coupling of β-arrestin1 in presence of overexpressed GRKs is, in fact, dependent on their kinase activity.

**Distinct AT1R–β-arrestin complex configurations are mediated by GRK2/3, GRK5/6 or PKC.** Multiple groups already investigated different AT1R–β-arrestin interactions[34,35]. However, we found that the GRK-specificity of β-arrestin complex configurations was even more intricate for the AT1R in comparison to the V2R. Therefore, we arranged the data obtained for the AT1R in a kinase-specific manner in Fig. 5a–f. We observed the most prominent

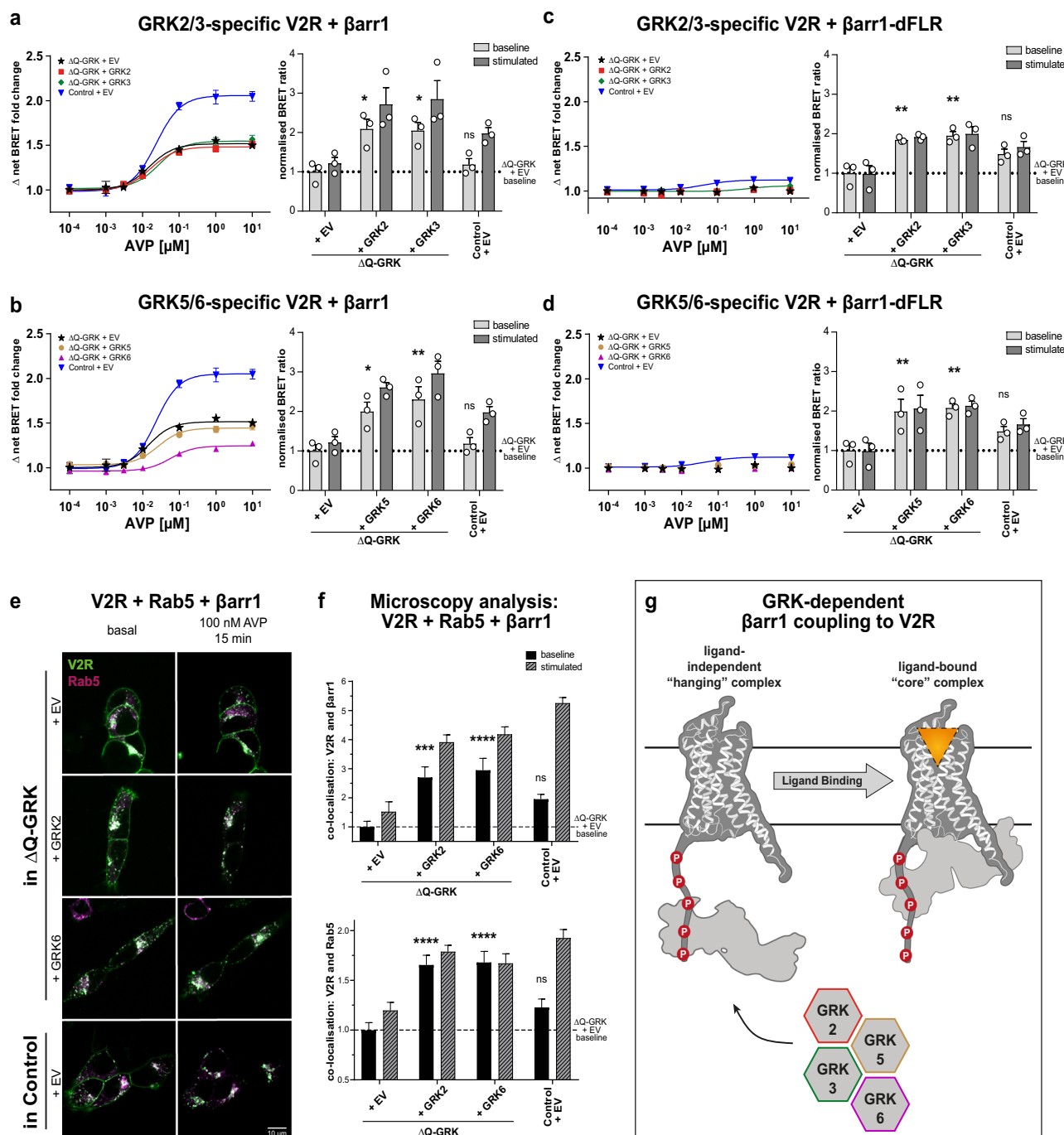

difference between angiotensin II (AngII)-induced β-arrestin1 and 2 recruitment to the AT1R, in ΔQ-GRK, as this condition features pronounced higher recruitment of β-arrestin2 (Supplementary Fig. 5g and Supplementary Fig. 6). Since the AT1R is a known target of heterologous desensitisation[30], we anticipated that PKC could be responsible for mediating this difference[35] at this Gq-coupled receptor. Hence, we conducted the experiment in the presence of Gö6983, a pan PKC inhibitor. This reduced the recruitment of both β-arrestins in Control cells and abolished GRK-independent β-arrestin1 recruitment (Supplementary Figs. 10a, b and 11a, b). In contrast, PKC inhibition had a negligible effect on the dynamic β-arrestin recruitment in the presence of overexpressed GRKs. Moreover, the overexpression of GRK5 and 6 showed lower dynamic recruitment of β-arrestin2 in comparison to ΔQ-GRK, regardless of PKC activity (Fig. 5e, f; Supplementary Fig. 10a, b). Taken together,

we conclude that dynamic GRK-mediated β-arrestin2 recruitment to the AT1R is unaffected by PKC activity, whereas for β-arrestin1 this is promoted by both GRKs and PKC.

Again, analysis of the measured BRET ratios reveals significantly increased basal molecular interaction in the presence of overexpressed GRK5 and 6 (Fig. 5e, f; Supplementary Fig. 6). This finding identifies the AT1R as yet another receptor that interacts with β-arrestins in a ligand-independent fashion, similar to the V2R. Interestingly, in this specific case, GRK5 and 6 seem to be able to account for the observed interactions. Notably, our statistical analysis shows a significant increase for the baseline of GRK3-mediated β-arrestin recruitment as well. However, since the greater part of the recruitment seems to be ligand-dependent in this condition, we did not conclude efficient AT1R–β-arrestin pre-coupling mediated by GRK3.

**Fig. 4 GRK2, 3, 5 or 6 can individually mediate a ligand-independent interaction of the V2R and β-arrestin1. a–d** ΔQ-GRK or Control cells were transfected with V2R-Halo-Tag and one of the following β-arrestin1 (βarr1)-NanoLuc fusion constructs: wild type (**a**, **b**) or βarr1 lacking the finger loop region (dFLR; **c**, **d**). Additionally, either GRK2, 3, 5, 6 or the empty vector (EV) were transfected as indicated. The dynamic BRET changes are shown as ligand concentration-response curves normalised to baseline values and vehicle control. All data points are calculated as Δ net BRET fold change, mean of $n = 3$ independent experiments ± SEM. The same dataset is presented as bar graphs, displaying the mean BRET values + SEM before (baseline) and after stimulation with 10 μM [Arg$^8$]-vasopressin (AVP; stimulated), normalised to the basal BRET ratio derived from the ΔQ-GRK + EV condition (dashed line). To test whether the baseline BRET ratios were significantly elevated compared to the respective ΔQ-GRK + EV baseline, an ANOVA and one-sided Dunnett's test was performed (*$p < 0.05$; **$p < 0.01$; ns not significant; **a**, **b** ΔQ-GRK + GRK2 $p = 0.0113$, ΔQ-GRK + GRK3 $p = 0.0142$, ΔQ-GRK + GRK5 $p = 0.0188$, ΔQ-GRK + GRK6 $p = 0.0036$, Control + EV $p = 0.6186$; **c**, **d**: ΔQ-GRK + GRK2 $p = 0.0073$, ΔQ-GRK + GRK3 $p = 0.0033$, ΔQ-GRK + GRK5 $p = 0.0024$, ΔQ-GRK + GRK6 $p = 0.0013$, Control + EV $p = 0.1054$). **e**, **f** ΔQ-GRK or Control cells were transfected with V2R-CFP (green), Rab5-mCherry (magenta), βarr1-YFP (not shown) and either EV, GRK2 or GRK6 as indicated. Images were taken before (basal) and after 15 min of 100 nM AVP stimulation. Representative images are shown in (**e**) and Supplementary Fig. 8e–h. The co-localisation of V2R and βarr1 or Rab5 was quantified using Squassh and SquasshAnalyst (number of images per respective condition; ΔQ-GRK + EV (35), ΔQ-GRK + GRK2 (35), ΔQ-GRK + GRK6 (33), Control + EV (30)). **f** Data are presented as mean fold change in co-localisation signal + SEM normalised to unstimulated (baseline) ΔQ-GRK + EV condition. Co-localisation prior to stimulation was compared using ANOVA and two-sided Dunnett's test (*$p < 0.05$; **$p < 0.01$; ***$p < 0.001$; ****$p < 0.0001$; ns not significant). All exact $p$ values, test statistics, effect sizes, confidence intervals and degrees of freedom are provided in the Source Data files. **g** Schematic depiction of βarr1 interactions with the V2R in the absence and presence of ligand, facilitated by high expression levels of GRKs.

After the discovery of these fundamental kinase-specific effects, we investigated whether this AT1R–β-arrestin complexes occur in a "hanging" or "core" configuration (Fig. 5; Supplementary Table 4). As stated above, PKC inhibition reduced AngII-induced β-arrestin2 recruitment in ΔQ-GRK, but the recruitment is still detectable. However, deletion of the FLR abolished AngII-induced β-arrestin2 recruitment in ΔQ-GRK independently of PKC activity (Fig. 5b). Thus, we conclude that the FLR is essential for the PKC-mediated AT1R–β-arrestin2 complex, as well as a GRK- and PKC-independent complex. This suggests that PKC activity alone cannot mediate a "hanging" complex configuration.

In the presence of overexpressed GRK2 or 3, dynamic, AngII-induced β-arrestin2 recruitment is neither altered by PKC inhibition nor deletion of the FLR (Fig. 5c, d). Interestingly, upon PKC inhibition the measured absolute BRET ratios are decreased to the levels recorded for the AngII-induced recruitment of the dFLR construct. This led us to the assumption that PKC inhibition and the deletion of the FLR mediate the same biological effect. Since PKC activity alone was shown to only mediate a "core" complex configuration (Fig. 5b) and the FLR is dispensable for GRK2/3-mediated, AngII-induced β-arrestin2 binding, we propose that GRK2 and 3 predominantly facilitate the formation of a "hanging" complex. Despite this, with our experimental setup, we cannot exclude the formation of a "core" complex between the two proteins under physiological conditions with unaltered PKC activity.

The pre-coupling effect mediated by GRK5 and 6 (Fig. 5e, f) is also observed for the β-arrestin2-dFLR mutant, suggesting the formation of a ligand-independent "hanging" complex with β-arrestin2. Again, the remaining ligand-dependent increase in β-arrestin2 recruitment measured for GRK5 overexpression could reflect ligand-activation of the pre-coupled "hanging" complex and subsequent formation of a tight "core" complex (Fig. 5g), similar to the mechanism proposed for the V2R (Fig. 4g). Notably, GRK6-mediated pre-coupling was reduced, indicating that the FLR plays a role in this process.

The utilisation of GRK KD mutants revealed that the pre-coupling of β-arrestin2 is mediated by GRK5 or 6 kinase activity (Supplementary Fig. 10c, e, g). Particularly, the phosphorylation of inactive AT1R in the presence of overexpressed GRK5 has been reported before[36]. Interestingly this β-arrestin2 pre-coupling effect was not observed in ΔGRK2/3/6 or ΔGRK2/3/5 (Supplementary Fig. 10d, f), indicating that it is dependent on individual GRK expression levels. This could explain how the same receptor might be differentially regulated in specific tissues, cellular compartments[9] or under pathophysiological conditions featuring dysregulated GRK expression levels[37].

In general, the GRK-dependent interaction of both β-arrestin isoforms and the AT1R is similar (Supplementary Fig. 11c–f). In the case of β-arrestin1, GRK5 and 6 overexpression also led to an enhanced basal interaction with AT1R (Supplementary Fig. 11g, h). Additionally, we conclude that both β-arrestins can use PKC phosphorylation to further stabilise a "core" complex with the AT1R. Interestingly, we observed recruitment of β-arrestin2 in absence of GRK and PKC phosphorylation, whereas β-arrestin1 does not seem to be able to interact with the unphosphorylated receptor. Thus, we can exclude the formation of a GRK- and PKC-independent "core" complex for β-arrestin1 (Fig. 5g, Supplementary Fig. 11).

**Ligand-independent AT1R regulation by GRK6 leads to receptor internalisation and impaired signalling responses.** To test if these different kinase effects have a direct impact on receptor functionality, we employed confocal microscopy and dynamic mass redistribution[38] (DMR) measurements. In ΔQ-GRK the AT1R did not show pronounced internalisation, while GRK2 overexpression strongly supports AngII-dependent receptor internalisation. In contrast, the AT1R was already found in intracellular compartments when overexpressing GRK6, independently of ligand application (Fig. 6a, b; Supplementary Fig. 12).

As expected, AngII evoked robust primary receptor signalling in GRK-deficient cells when assessed by DMR (Fig. 6c, d). Interestingly, co-transfection of ΔQ-GRK with β-arrestin2 did not suffice to diminish cellular AT1R signalling despite significant β-arrestin2 recruitment under comparable experimental conditions (Fig. 6a compare with 5b). Even though PKC-specific phosphorylation can stabilise AT1R–β-arrestin2 interactions (Fig. 5b), we conclude that GRK phosphorylation is strictly required for β-arrestin2-mediated receptor desensitisation and internalisation (Fig. 6b). The cellular signalling responses were significantly dampened under GRK2 overexpression, confirming efficient receptor desensitisation in this condition. In this system GRK2 functions as a canonical sensor for receptor activation, governing location and activity of GPCRs via the mediation of β-arrestin functions.

The overexpression of GRK6 almost eliminated the measured cellular signalling response (Fig. 6c, d). This, in combination with the observed pattern of AT1R subcellular localisation, suggests that GRK6-mediated AT1R phosphorylation precludes a majority of receptor molecules from the membrane and thus from being exposed to the ligand. This is especially significant, as it demonstrates that the upregulation of GRKs could have two distinctly different consequences. Depending on the combination of involved kinases and receptors, it could either lead to a canonical

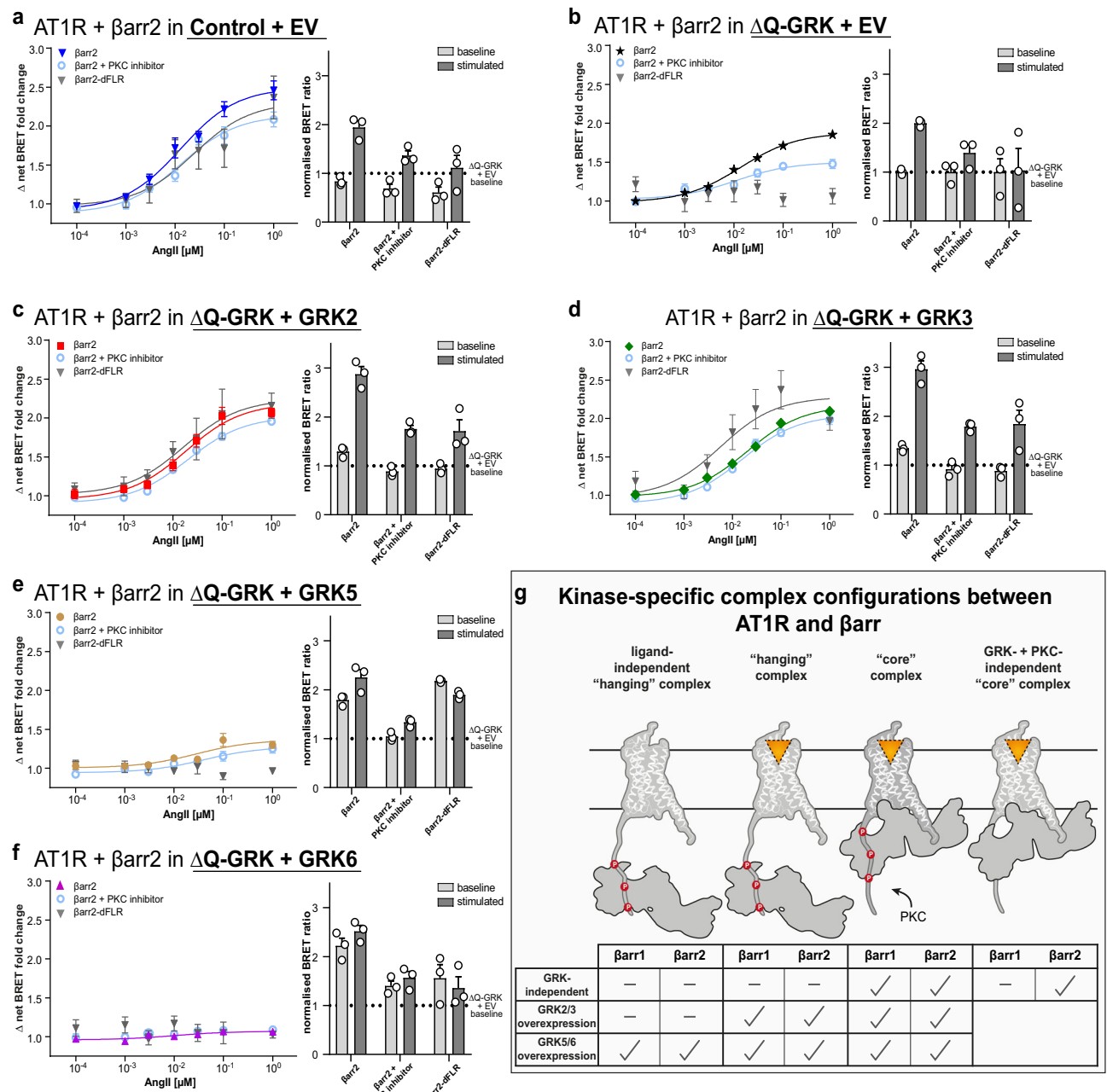

**Fig. 5 Distinct AT1R–β-arrestin2 complex configurations are mediated by GRK2/3, GRK5/6 or PKC. a–f** To identify the contribution of the individual GRKs to the complex formation of β-arrestin2 (βarr2) with AT1R, Control or ΔQ-GRK cells were transfected with AT1R-NanoLuc, Halo-Tag-βarr2 and either GRK2, 3, 5, 6 or the empty vector (EV) as indicated, in absence or presence of PKC inhibitor Gö6983 (500 nM). Additionally, the GRK-specific βarr2 recruitment to the AT1R was measured utilising a Halo-Tag-βarr2 construct lacking the finger loop region (dFLR). Angiotensin II (AngII)-induced dynamic BRET changes are shown as concentration-response curves. All data points are calculated as Δ net BRET fold change normalised to baseline values and vehicle control, represented as the mean of $n = 3$ independent experiments ± SEM. The data are also presented in respective bar graphs, displaying the mean BRET values + SEM before (baseline) and after stimulation with 1 µM AngII (stimulated), normalised to the basal BRET ratio derived from the corresponding ΔQ-GRK + EV condition in (**b**). The results of the statistical analysis of displayed data are listed in Supplementary Table 4. All test statistics, effect sizes, confidence intervals and degrees of freedom are provided in the Source Data files. **g** Schematic summary of the kinase-specific complex configurations between AT1R and βarr1 or 2 either in the absence of GRKs, mediated by GRK2/3 or GRK5/6 overexpression as observed in (**a–f**; Supplementary Fig. 11).

increase in efficiency to induce arrestin-mediated receptor regulation or an almost complete loss of GPCR responsiveness.

## Discussion

The establishment of various GRK KO cell lines enabled us to identify biological patterns of GRK-specific β-arrestin-mediated

GPCR regulation. Our comprehensive analysis revealed clustering of GPCRs into different groups, i. a. GRK2/3-regulated and GRK2/3/5/6-regulated receptors. While the V2R and AT1R are both regulated by all tested GRKs, they exhibited substantial β-arrestin pre-coupling upon GRK overexpression and therefore constitute a subgroup of GRK2/3/5/6-regulated receptors. Another conclusion that can be drawn from the presented GRK-

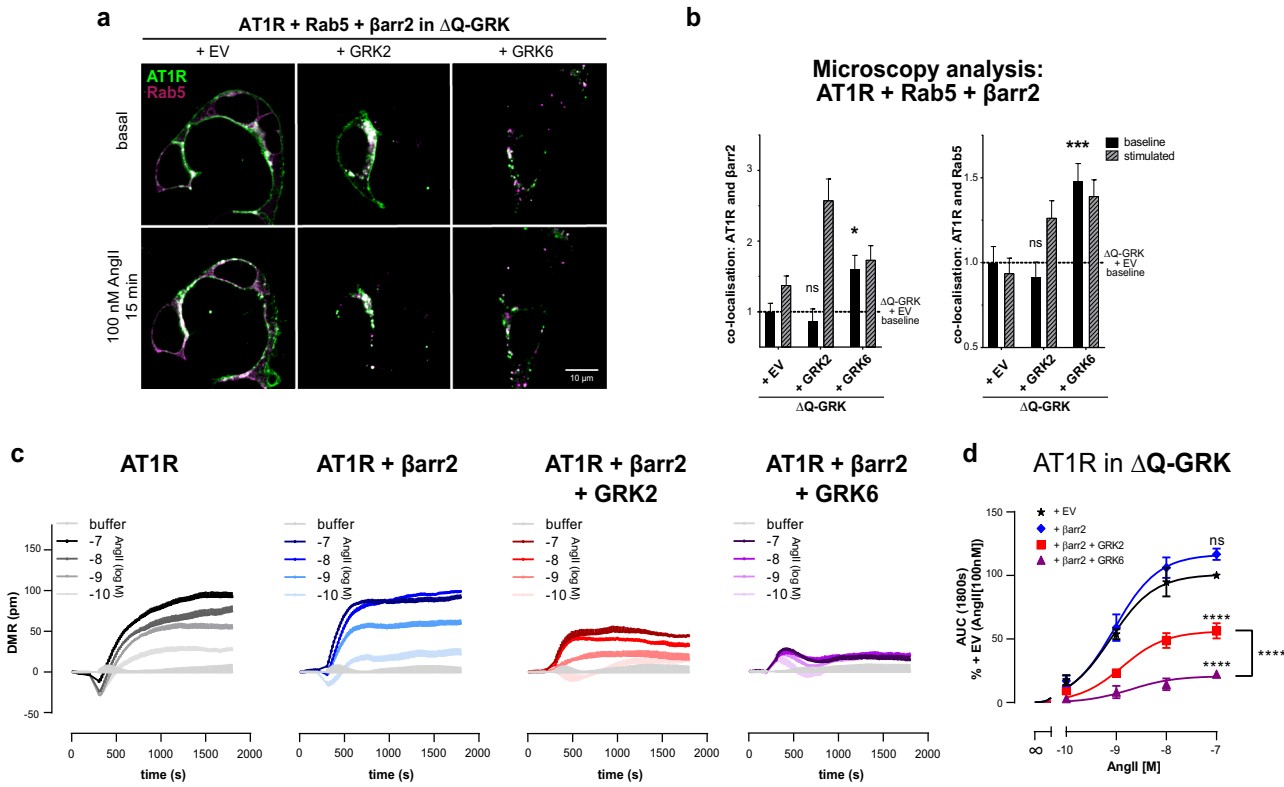

**Fig. 6 Pre-coupling of β-arrestin2 mediated by overexpression of GRK6 results in desensitisation and constitutive internalisation of the AT1R.**
**a**, **b** ΔQ-GRK cells were transfected with hsp-AT1R-CFP (green), Rab5-mCherry (magenta), βarr2-YFP (not shown) and either EV, GRK2 or GRK6, as noted. Images were taken before (basal) and after 15 min of 100 nM AngII stimulation. Representative images are shown in (**a**) and Supplementary Fig. 12. The co-localisation of AT1R and βarr2 or Rab5 was quantified using Squassh and SquasshAnalyst (number of images per respective condition; ΔQ-GRK + EV (31), ΔQ-GRK + GRK2 (31), ΔQ-GRK + GRK6 (33)) (**b**). Data are presented as mean fold change in co-localisation signal + SEM normalised to unstimulated ΔQ-GRK + EV. Co-localisation prior to stimulation was compared using ANOVA and two-sided Dunnett's test (*$p < 0.05$; **$p < 0.01$; ***$p < 0.001$; ns not significant; basal AT1R co-localisation with βarr2: ΔQ-GRK + GRK2 $p = 0.812$, ΔQ-GRK + GRK6 $p = 0.027$; basal AT1R co-localisation with Rab5: ΔQ-GRK + GRK2 $p = 0.7649$, ΔQ-GRK + GRK6 $p = 0.0008$). **c** ΔQ-GRK cells transiently transfected with hsp-AT1R-CFP, with or without co-transfection of either βarr2 alone or in combination with GRK2 or GRK6, were stimulated with AngII and real-time dynamic mass redistribution (DMR) responses were recorded as a measure of AT1R activity. DMR recordings are shown as mean + SEM of three technical replicates from a single experiment, representative of $n = 3$ independent experiments. **d** Concentration-effect curves derived from $n = 3$ independent experiments (representatively shown in (**c**)) ± SEM are plotted from the area under the curve (AUC) within 0 and 1800 s. Data are normalised to the picometre wavelength shifts evoked with 100 nM AngII in cells expressing hsp-AT1R-CFP alone. Statistical significance was calculated using a two-way ANOVA and two-sided Tukey's test (*$p < 0.05$; **$p < 0.01$; ***$p < 0.001$; ****$p < 0.0001$; ns not significant). All exact $p$ values, test statistics, effect sizes, confidence intervals and degrees of freedom are provided in the Source Data files.

specific β-arrestin recruitment screen (Supplementary Fig. 5) is the increased ability of β-arrestin2 to form GRK-independent complexes with GPCRs, as compared to β-arrestin1. Multiple experiments in this study suggest that β-arrestin2 exhibits higher recruitment in ΔQ-GRK than β-arrestin1, regardless of which GPCR was tested. This behaviour of β-arrestin2 was already hypothesised in the publication of Zhan et al.[39]. The authors provided the crystal structure of β-arrestin2 and showed that the protein displays higher flexibility than other arrestin isoforms. Because of this, β-arrestin2 is more likely to "probe" different inactive and active conformations, even in the absence of phosphorylated and/or active GPCRs. This property of β-arrestin2 is attributed to a more disordered C-domain, as compared to other arrestins. Specifically, β-sheet XIV (located in the C-domain β-sandwich) appears to be shortened, in comparison to β-arrestin1, and continues as an unstructured loop. As this structural component also takes part in interactions of the arrestin hinge region, close to the polar core, structural instability could lead to the spontaneous activation of β-arrestin2, without the need to engage phosphorylated intracellular domains of GPCRs. Thus, the authors of Zhan et al.[39] provide an adequate explanation of why

β-arrestin2 might be better suited to form phosphorylation-independent GPCR-complexes. Notably, this hypothesis has also been tested and supported by modelling[40].

Strikingly, we were able to show that certain GPCRs are readily being regulated by overexpressed GRKs in a cellular system without ligand addition. Although ligand-independent GPCR phosphorylation has been described for multiple receptors[36,41–44], it has not been convincingly shown to this point that this phosphorylation would translate into arrestin functions in a cellular context. We demonstrated that both, the V2R and AT1R couple to arrestins and internalise in a ligand-independent fashion as long as the essential GRKs are present at high expression levels (Figs. 4–6). Although GRK5 and 6 mediate this β-arrestin pre-coupling for both receptors, this is not a unique feature of the membrane-associated GRK4 family kinases. As GRK2 and 3 can achieve similar effects for the V2R, this rather has to be a distinct characteristic of a specific GPCR. Furthermore, the significantly increased basal molecular interaction between β-arrestin2 and the respective receptor upon GRK overexpression was not abolished by pre-treatment with an inverse agonist or antagonist (Losartan[45] was used for the AT1R and Tolvaptan[46] for the V2R, Supplementary Figure 13). Hence,

this specific pre-coupling effect is unlikely due to the constitutive activity of the respective GPCR. Especially, since this pre-coupling effect could not be transferred to the b2AR by exchange with the V2R C-terminus (Supplementary Figure 5b, c, m; 6), we can also exclude the C-terminus as sole mediator of GRK-specific processes.

Currently, it is unknown whether a single receptor is phosphorylated by a single kinase or by multiple kinases in a sequential manner. This could lead to vastly different outcomes, as we were able to show that different GRKs and second messenger kinases are able to induce divergent regulatory processes, depending on the targeted GPCR. Notably, the effect of PKC on the regulation of GRKs was described multiple times in literature[47–49], as it was shown that the activity of GRKs can be either increased or decreased via PKC activity, depending on the used system. Even though we were able to record a reduction in GRK5- and 6-mediated β-arrestin recruitment to the AT1R under inhibition of PKC (Fig. 5e, f), we could not draw a conclusion on how PKC modulates the activity of specific kinases. The elucidation of these intriguing effects requires more in-depth analysis and possibly single-molecule studies.

Using our triple GRK KO cell lines, it became evident that different GPCRs require certain levels of GRK expression in order to recruit arrestins. In contrast to the b2AR (Fig. 1c), the PTH1R and M5R showed robust β-arrestin2 recruitment in ΔGRK3/5/6 cells (Supplementary Fig. 7a, b). This does not reflect on the ability of GRK2 to regulate the b2AR, as has been shown multiple times in literature and our presented overexpression experiments (Fig. 1e). These results rather demonstrate that the affinities of GRK isoforms to GPCRs differ depending on the individual receptor. Thus, the tissue-specific expression levels[50] of individual GRKs in combination with their affinities to or formed complex configurations[51] with specific receptors determine GPCR regulation.

Several studies have demonstrated phosphorylation of the b2AR at different serine and/or threonine residues[43,52] and phosphorylation by GRK2 or 6 was shown to serve different functions[8]. In our study, we observed that all GRKs can mediate receptor–β-arrestin interactions to the same extent and by using a single β-arrestin2 conformational change sensor (FlAsH5 according to Nuber et al.[28]), we could demonstrate that the N-domain of β-arrestin2, which recognises phosphorylated intracellular receptor domains, showed similar conformational changes for the different kinases (Fig. 2a). However, more experiments have to be performed to rule out that differential b2AR phosphorylation by GRK2 or 6 might lead to distinct β-arrestin2 conformational changes. While different GRK isoforms might preferably phosphorylate distinct sites of the b2AR, resulting in different phosphorylation patterns of the C-terminus or IL3, our experiments clarify that the phosphorylation by each GRK isoform is sufficient to induce high-affinity β-arrestin recruitment (Fig. 1e).

Interactions between arrestins and the M2R were investigated previously[53–55]. Interestingly, the β-arrestin recruitment assay only induced minimal BRET changes for the M2R at endogenous GRK expression levels (Supplementary Fig. 14). However, upon overexpression of GRK2 or 3 robust β-arrestin recruitment was observed. It is tempting to speculate that the M2R exhibits a rather low affinity for GRKs to prevent its desensitisation since its function is essential for the reduction of heart rate[56]. Under pathophysiological conditions of GRK2 overexpression during chronic heart failure[57], the M2R might internalise which could possibly contribute to tachycardiac effects in patients.

For each receptor case with unclear GRK assignment by statistical analysis (Fig. 3i, j), alternative kinases were previously reported to be involved in receptor phosphorylation. In the case of the M1R and M3R, casein kinase 1 alpha and casein kinase 2 were shown to be involved in receptor phosphorylation, respectively[58,59]. For the C5aR1, PKCβ was shown to contribute

to receptor phosphorylation[60]. Therefore, our cellular platform for arrestin recruitment might be able to rapidly differentiate between receptors with purely GRK-dependent arrestin recruitment and receptors that rely on the action of other intracellular kinases for efficient arrestin binding, desensitisation and internalisation.

Using our ΔQ-GRK cell line, we were able to show that high expression levels of GRK2 and 3 are able to mediate β-arrestin interactions with all tested GPCRs. GRK5 and 6 seem to fulfil divergent roles depending on the analysed GPCR, as they were not able to induce β-arrestin-coupling to the M2R, M4R, M5R and MOP (Fig. 3i, j; Supplementary Fig. 5). Interestingly, this is not necessarily due to a lack of receptor phosphorylation, as we were able to show that GRK5 and 6 phosphorylate the MOP upon agonist activation, but fail to mediate β-arrestin recruitment and receptor internalisation (Supplementary Fig. 2). Nevertheless, we found receptors for which the GRK5- and 6-facilitated receptor regulation is indistinguishable from that mediated by GRK2 and 3 (namely the PTH1R and b2AR, Figs. 1e; 3c, d).

In an endeavour to match the measured GRK-specific β-arrestin recruitment with the main features of the tested GPCRs, we analysed the length of the respective C-terminus and IL3, as well as the number and relative location of their putative phosphorylation sites (Fig. 7, Supplementary Table 2). Here we were not able to find correlations that would compellingly explain the observed GRK-selectivity for our panel of GPCRs (Fig. 7a). Interestingly, none of the analysed class B receptors are solely regulated by GRK2 and 3, but all are regulated by GRK2/3/5/6 (b2V2 was excluded from this analysis as an unphysiological chimaera; Fig. 7b).

Further, we assessed the abundance and relative positions of previously established phosphorylation motifs[61] (PPP, PXPP, PXPXXP, PXXPXXP, whereby P represents either Ser, Thr, Asp or Glu residues and X any amino acid), reported to be important for β-arrestin recruitment[62,63]. As illustrated in Fig. 7c–n, all motifs show a similar distribution between our defined GRK-selectivity groups.

Interestingly, the PXPP motif showed a higher abundance in the central area (0.25–0.75) of analysed peptide stretches of GPCRs regulated by GRK2/3. In contrast, the same motif was found more often in the peripheral area (0.00–0.25 and 0.75–1.00) of peptide stretches of GPCRs regulated by GRK2/3/5/6 (Fig. 7h). We found a significant association (Fisher's exact test, $p = 0.0005$) between the position of putative phosphorylation motifs (PXPP: central vs. peripheral) and GRK-specificity of the assessed GPCRs, according to our analysis. However, the causality of this association remains to be explained. Notably, although class A and B GPCRs are differentially represented in the GRK-specificity groups as defined in Fig. 3i, j, no significant association was found between the positions of PXPP motifs and the A–B classification (Fig. 7l; Fisher's exact test, $p = 0.7328$). In general, we did not identify common features of C-terminal and IL3 sequences which would allow for the reliable prediction of GRK-selectivity. This again underlines the complexity of the GPCR regulatory system, as we were not able to link the existence or positioning of phosphorylation patterns that are suitable for the recruitment of β-arrestins with the observed GRK selectivity. Hence, we propose that GRK-selectivity is rather defined by the overall geometry of each GPCR and influenced not only by the availability of putative phosphorylation sites but also by the general promiscuity of other intracellular domains.

Recently, it has been reported that phosphorylation patterns, which promote β-arrestin recruitment, do not coincide with phosphorylation sites promoting β-arrestin activation[64]. Hence, it is still possible that the herein identified GRK-selectivity for β-arrestin recruitment might differ from GRK-selectivity toward

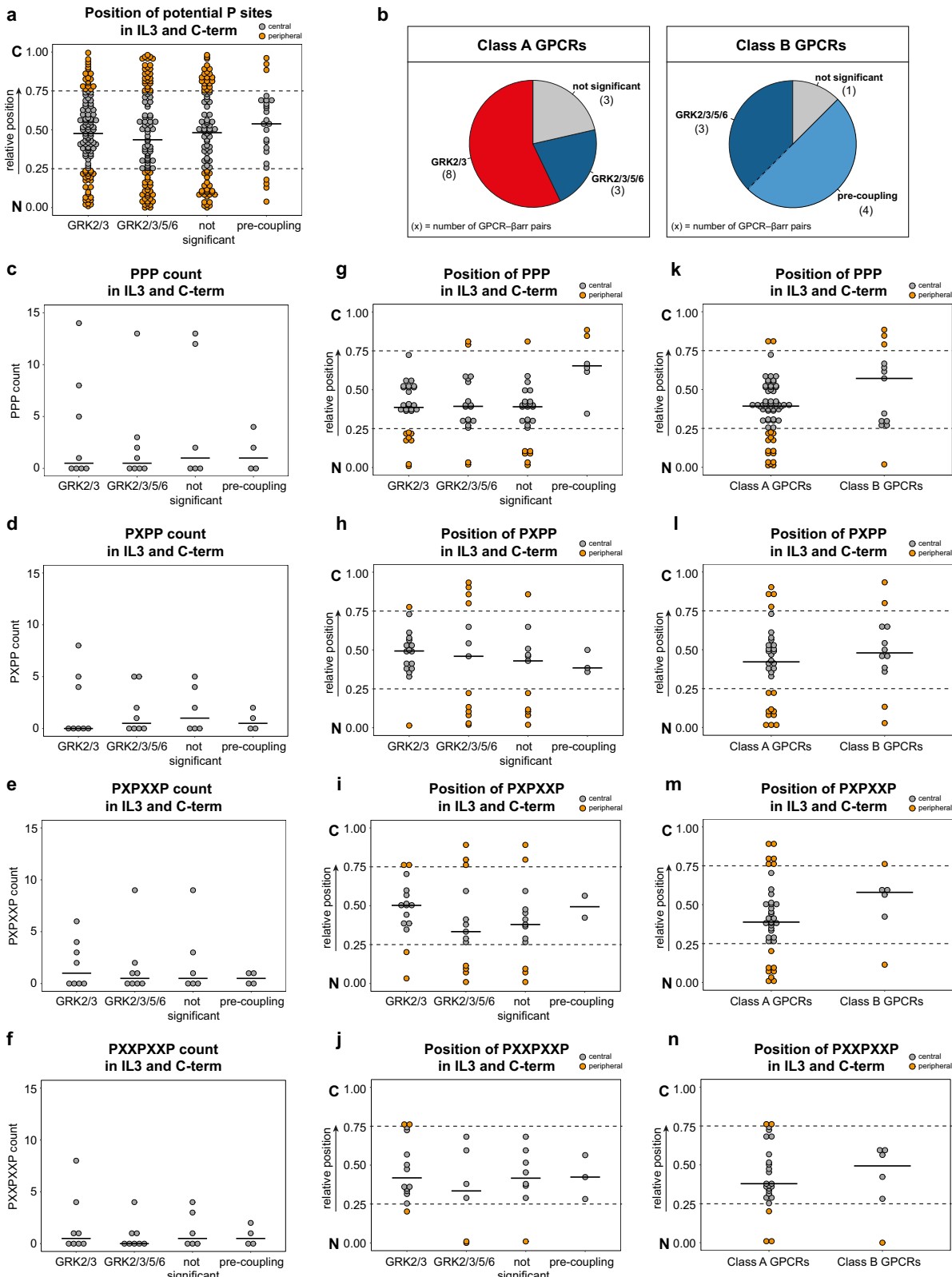

β-arrestin activation. More work has to be done to connect functional β-arrestin conformational changes with GRK-specific GPCR phosphorylation patterns to unravel those fine details of the phosphorylation barcode hypothesis.

Apart from phosphorylation patterns, Mayer et al.[65] proposed phosphorylation sites with specific functions based on conformational changes in arrestin-1 induced by binding to phosphopeptides. Intriguingly, the existence of inhibitory sites, which prevent arrestin binding, were predicted for several GPCRs including the AT1R. Since GRK5 and GRK6 overexpression abolished concentration-dependent β-arrestin recruitment to AT1R (Fig. 5e, f; Supplementary Fig. 11g, h), we created AT1R constructs lacking one of each inhibitory site (AT1R[S347A] and AT1R[S348A]) or both (AT1R[S347A/S348A]; Supplementary Fig. 15).

**Fig. 7 Putative phosphorylation motifs in IL3 and C-terminus of GPCRs are not associated with GRK-specific β-arrestin recruitment.** Analysis of the abundance and position of identified putative serine and threonine (Ser/Thr) phosphorylation sites, clusters[62] (PPP, PXPP) and patterns (PXXPXXP, PXPXXP, as identified by Zhou et al.[61]) in the intracellular loop 3 (IL3) and C-terminus (C-term; information from GPCRdb.org) of GPCRs listed in Supplementary Table 2 with the exception of the unphysiological b2V2 chimaera receptor. While X represents any amino acid, P may be a Ser, Thr or a negatively charged amino acid (glutamic acid, Glu; aspartic acid, Asp). The number and positions of potential phosphorylation sites, clusters and patterns were detected using Python 3.8.7. In order to compare positions of potential phosphorylation sites or motifs between GPCRs with varying lengths of IL3 and C-term, their relative position was calculated as the position index in relation to the full length of the respective peptide stretch. Consequently, the relative position of 0 corresponds to the beginning (N), whereas a relative position of 1 corresponds to the end (C) of the respective peptide stretch. All relative positions are displayed as dot plots with the corresponding median, indicated by bars. Positions between 0.25 and 0.75 were categorised as central (grey) while positions between 0.0 and 0.25 as well as 0.75 and 1.00 were considered peripheral (orange). Dashed lines mark 0.25 and 0.75 breakpoints. **a** Relative position of potential phosphorylation sites (Ser/Thr) in the IL3 and C-term of analysed GPCRs grouped according to their GRK-specific β-arrestin (βarr) recruitment. In addition to the three groups defined in Fig. 3i, j (GRK2/3-regulated, GRK2/3/5/6-regulated and not significant), GPCRs displaying βarr pre-coupling (AT1R and V2R) were compiled in a fourth group (pre-coupling). **b** The analysed receptors were grouped as class A or B according to Oakley et al.[62] (Supplementary Table 2) and each class is represented as one pie chart. In each pie chart, all GPCR–βarr pairs are assigned to the different GRK-specific βarr recruitment groups. The absolute number of GPCR–βarr pairs presented in each group is indicated in brackets. Since GPCRs displaying βarr pre-coupling can be considered a subgroup of GRK2/3/5/6-regulated GPCRs, both slices are separated by a dashed line (right pie). **c–f** Abundance of PPP clusters (**c**), PXPP clusters (**d**), PXPXXP patterns (**e**) and PXXPXXP patterns (**f**) are displayed as dot plots grouped according to their GRK-specific βarr recruitment. **g–n** Relative positions of P-patterns as indicated. They were grouped according to their GRK-specific βarr recruitment (**g–j**), as well as the GPCR classification (**k–n**). Association between the position of PXPP and the GRK-specific βarr recruitment (**h**, $p = 0.0005$) or GPCR classification (**l**, $p = 0.7328$) was tested using Fisher's exact test. Exact odds ratios and confidence intervals are provided in the Source Data files.

Neither of the three mutant receptors showed an altered β-arrestin2 pre-coupling upon GRK5 or 6 overexpression (Supplementary Fig. 15f–m), yet alanine substitution of the $S^{347}$ phosphorylation site reduced GRK2- and 3-specific β-arrestin2 recruitment to the level of PKC-mediated β-arrestin2 recruitment (Supplementary Fig. 15c). Thus, these two phosphorylation sites do not exhibit an inhibitory effect on β-arrestin2 recruitment. Although the effect on β-arrestin1 recruitment remains to be examined, our results highlight the difference between the cellular functionality of arrestins and their behaviour towards synthesised phosphopeptides. Hence, it is imperative that future studies take all interaction interfaces between GPCRs and β-arrestin into account, as our analysis shows that there are more determinants for β-arrestin functions besides receptor phosphorylation at the C-terminus or IL3.

While GRK5 and 6 are membrane-localised[66], GRK2 and 3 are primarily cytosolic and translocate to the plasma membrane supported by interactions with βγ-subunits of activated G proteins[67]. It is still conceivable that GRK5 and 6 interactions with certain GPCRs might be obstructed due to their distinct cellular localisation. The plasma membrane features a rather heterogeneous distribution of proteins and it has been shown that certain GPCRs tend to reside in specific membranous microdomains[68]. Some receptors might localise in membranous compartments that are inaccessible for GRK5 and 6. Following this hypothesis, these GPCRs would be accessible to GRK2 and 3 since they emerge from the cytosol and would not be limited to two-dimensional diffusion and hindered by possible confinements. This still does not exclude the existence of GPCRs which do not serve as substrates for GRK2 or 3, due to e.g. low affinity.

While this manuscript was under revision, another group published three independently created ΔGRK knockout cell lines (ΔGRK2/3, ΔGRK5/6 and ΔGRK2/3/5/6)[69], which correspond to our GRK family knockout cell lines and ΔQ-GRK. The only common receptor among both manuscripts is the C5aR1. Similar to our findings, they showed a stronger loss of β-arrestin1 recruitment to the C5aR1 in the absence of GRKs, as compared to β-arrestin2. Interestingly, using their family knockout cell lines, the authors found a very mild loss of arrestin recruitment for the knockout of GRK2 and 3, while the loss of GRK5 and 6 almost resembled the ΔGRK2/3/5/6 condition. This would imply that the C5aR1 is mainly phosphorylated by GRK5 and 6. In contrast, our overexpression experiments suggest that all four GRKs are able to facilitate β-arrestin recruitment to a similar extent. Only using

endogenous expression levels to investigate the influence of GRKs on β-arrestin recruitment could be misleading in this context, as this is strongly dependent on the utilised cell line. This is exemplified by the data shown in Fig. 1c, e. Here, we also determined GRK6 as the main driver of arrestin recruitment to the b2AR under endogenous expression levels of GRKs in HEK293 cells, while all GRKs are able to equally mediate this interaction when overexpressed. This strengthens the argument that GRK expression levels strongly influence the GRK-specificity of β-arrestin recruitment[50].

In conclusion, we were able to elucidate the GRK-specificity of receptor regulation for 12 different GPCRs. Our analysis demonstrates that different GRK isoforms may have identical, overlapping or divergent functions, depending on the targeted GPCR. This adds another layer of complexity to the regulation of GPCR signalling and trafficking and a possible explanation of how different β-arrestin functions are mediated across various tissues and cell types, especially considering often dysregulated, pathophysiological GRK expression levels.

## Methods

**Cell culture.** HEK293 cells were originally obtained from DSMZ Germany (ACC 305) and cultured in Dulbecco's modified Eagle's medium (DMEM; Sigma-Aldrich D6429), complemented with 10% foetal calf serum (Sigma-Aldrich F7524) and 1% of penicillin and streptomycin mixture (Sigma-Aldrich P0781) at 37 °C with 5% $CO_2$. The cells were passaged every 3–4 days. Cells were regularly checked for mycoplasma infections using the LONZA MycoAlert mycoplasma detection kit (LT07-318).

**CRISPR/Cas 9 mediated knockout of GRK2, 3, 5 and 6.** Stable GRK knockout cells were generated by transient transfection using self-made PEI reagent (Sigma-Aldrich, 408727, diluted to 10 μg/ml, pH 7.2, adjusted with HCl) of the parental cells (HEK293) with lentiCRISPR v2 plasmid[70] (Addgene #52961) containing target-specific gRNAs listed in Supplementary Table 5. Complementary forward and reverse oligos were annealed and ligated into the BsmBI-restricted lenti-CRISPR v2 vector. This vector could also be used for the generation of viral particles, but in our approach, they were directly transfected into the target cells. In order to prevent side effects caused by multiple transfections and selection rounds, all cell clones were created in singular attempts. To knockout one specific GRK, four different gRNA constructs were simultaneously transfected. Inline, double, triple or quadruple knockout cells were generated by transfection of 8, 12 or 16 respective gRNA constructs at once. The transfected cells were then selected using 1 μg/ml puromycin (Sigma-Aldrich #P8833). Limited dilution was used to establish single-cell clones, which were then analysed for the absence of the target protein by Western blot analysis. A puromycin selected cell pool transfected with empty lentiCRISPR v2 plasmid was used as Control.

**Establishment of GRK expression constructs**. The used pcDNA3-GRK2 expression construct was described before[71]. GRK3 (NCBI reference sequence NM_005160.4), GRK5 (NCBI reference sequence NM_005308.3) and GRK6 (NCBI reference sequence NM_001004106.3) were amplified by PCR using GRK-specific primers including restriction sites for HindIII (forward primer) and BamHI (reverse primer) (GRK3: Forward primer (fw)—CTT AAG CTT GCC ACC ATG GCG GAC CTG GAG GCTG, Reverse Primer (rev)—CTT AGG ATC CTA GAG GCC GTT GCT GTT TCTG; GRK5: fw—CTT AAG CTT GCC ACC ATG GAG CTG GAA AAC ATC GTG, rev—CTT AGG ATC CTA GCT GCT TCC GGT GGAG; GRK6: fw—CTT AAG CTT GCC ACC ATG GAG CTC GAG AAC ATC GTAG, rev—CTT AGG ATC CTA GAG GCG GGT GGG GAGC). GRK3 and 6 were amplified from human leucocyte cDNA, GRK5 was amplified from a beta-galactosidase fusion plasmid described before[72]. The fragments were ligated into pcDNA3 plasmids after BamHI and HindIII digest. Sequences of all plasmids were validated by sequencing.

The kinase-dead (KD) mutants of GRK2, GRK5 and 6 were created by site-directed mutagenesis resulting in GRK2-K220R, GRK5-K215R and GRK6-K215R.

**Western blot**. Cells were washed once with ice-cold PBS and subsequently lysed with RIPA Buffer (1% NP-40, 1 mM EDTA, 50 mM Tris pH 7.4, 150 mM NaCl, 0.25% sodium deoxycholate), supplemented with protease and phosphatase inhibitor cocktails (Roche, #04693132001, #04906845001). Cleared lysates were boiled with sodium dodecyl sulfate (SDS) loading buffer and 15 μg of total protein were loaded onto each lane of 10% polyacrylamide gels. After transfer onto nitrocellulose membranes, the total protein was detected by using specific antibodies (GRK2: Santa Cruz sc-13143 (1:500); GRK3: Cell signalling technology #80362 (1:250); GRK5: Santa Cruz, sc-518005 (1:250); GRK6: Cell signalling technology #5878 (1:1000), Vinculin: Biozol BZL03106 (1:1000): Actin: Sigma-Aldrich, A5441 (1:2000)). As secondary antibodies, we used SeraCare peroxidase-conjugated, Goat anti-rabbit (No. 5220-0336) and Goat anti-mouse (No. 5220-0341), 1:10,000.

For experiments using the MOP, phospho-specific antibodies see below. Quantification of the blots was done using Fujifilm Multi Gauge Software (V3.0).

Relative quantification of endogenous GRK expression in Control cells was performed as described elsewhere[27]. In brief, ΔQ-GRK cells were transfected with N-terminally HA-tagged GRK2, 3, 5 and 6 constructs and lysed after 24 h. Protein standards of equal amounts of HA-tagged GRKs were prepared and loaded onto the same gel with three different lysates from Control cells. Three independent blots of the same lysates and standards were prepared and probed with the GRK specific antibodies. After quantification, the relative protein abundance was calculated in relation to the signal of the respective standard. The GRK3 signal detected by the GRK2 antibody and GRK5 signal detected by the GRK6 antibody was not considered.

**Cell viability assay and determination of growth rates**. Cells were seeded with a density of 5000 cells/100 μl in 96-well plates, with 12–16 wells per cell line. After 48 h of incubation at 37 °C and 5% CO$_2$, 20 μl of the cell titre blue reagent (Promega, G8081) were added to 3–4 wells for each cell line, and the cells were further incubated for 1.5 h under the same growth conditions. After the incubation time, fluorescence (excitation 540 nm, emission 610 nm, Gain 40, top reading) was measured using a TECAN Infinite 200 (Tecan, Crailsheim, Germany) plate reader. The plate was further incubated and 72, 96 and 120 h after seeding, three wells were measured as just described. The proliferation was reported as a relative fluorescence signal compared to the first measurement (48 h after seeding) ±standard error of the mean (SEM) of at least $n = 3$ experiments. Using these data, the growth rates were determined for the three 24 h intervals and mean values ± SEM are reported.

**Stable MOP-expressing cells and immunoprecipitation experiments**. A retroviral expression vector was created by replacing the mCherry expression cassette of pMSCV-IRES-mCherry FP (a gift from Dario Vignali, Addgene plasmid #52114) with a neomycin resistance cassette of pcDNA3 plasmid resulting in an empty pMSCV-IRES-NEO vector. The open reading frame of N-terminal haemagglutinin (HA) tagged murine MOP gene[72] was inserted into the multi-cloning-site of this vector resulting in pMSCV-HA-MOP-IRES-NEO. Freshly produced retroviral particles were used to transduce GRK-KO cell clones or respective Control cells. The cells were selected with 1 mg/ml G418 (Gibco, 11811-031) for 10 days.

For immunoprecipitation experiments, $4 \times 10^6$ cells were seeded in a 21 cm$^2$ dish and after 24 h stimulated for ten minutes with 10 μM DAMGO ([D-Ala2, N-MePhe4, Gly-ol]-enkephalin, Tocris 1171) in DMEM medium (Sigma-Aldrich, D6429) without supplements. After washing with ice-cold PBS, the cells were lysed in 500 μl RIPA buffer as described in the "Western blot" section. Totally, 400 μg of total protein lysates were incubated with 20 μl of HA-beads slurry (Thermo Scientific, 26182) at 4 °C on a turning wheel for 2 h. The beads were then washed three times with RIPA buffer containing protease and phosphatase inhibitors and 75 μl sample buffer (125 mM Tris pH 6.8, 4% SDS, 10% glycerol, 167 mM DTT) was added and the samples were heated to 42 °C for 20 min. The samples were loaded onto 10 % polyacrylamide gels (7.5 μl per 15-well mini gel lane) and phosphorylation was detected using freshly prepared solution of rabbit polyclonal phosphosite-specific MOP antibodies[22,23] (dilution for all 1:1000) anti-pT370 (7TM0319B), anti-pS375 (7TM0319C) anti-pT376 (7TM0319D) and anti-pT379

(7TM0319E), all obtained from 7TM Antibodies (Jena, Germany). The total receptors were detected with an anti-HA-antibody (Cell signalling technology # 3724; 1:1000). Quantification of the blots was done using Fujifilm Multi Gauge Software (V3.0).

**Intermolecular bioluminescence resonance energy transfer (BRET)**. The GRK-selective β-arrestin recruitment assay was performed either in Control or specific ΔGRK cell lines. In 21 cm$^2$ dishes, $1.6 \times 10^6$ cells were seeded and transfected the next day with 0.5 μg of the respective GPCR C-terminally fused to Nano luciferase (NanoLuc), 1 μg of β-arrestin constructs N-terminally fused to a Halo-ligand binding Halo-Tag and 0.25 μg of one GRK or empty vector. In the case of the PTH1R and V2R, the BRET pair was swapped. All transfections were conducted following the Effectene transfection reagent manual by Qiagen (#301427) and then incubated at 37 °C overnight. Into poly-D-lysine-coated 96-well plates (Brand, 781965), 40,000 cells were seeded per well in presence of Halo-ligand (Promega, G980A) at a ratio of 1:2000. A mock labelling condition without the addition of the Halo-ligand was seeded for each transfection. After 24 h, the cells were washed twice with measuring buffer (140 mM NaCl, 10 mM HEPES, 5.4 mM KCl, 2 mM CaCl$_2$, 1 mM MgCl$_2$; pH 7.3) and NanoLuc-substrate furimazine (Promega, N157B) was added in a ratio of 1:35,000 in measuring buffer. A Synergy Neo2 plate reader (Biotek), operated with the Gen5 software (version 2.09), with a custom-made filter (excitation bandwidth 541–550 nm, emission 560–595 nm, fluorescence filter 620/15 nm) was used to perform the measurements. The baseline was monitored for 3 min. After the addition of the respective agonist, the measurements were continued for five minutes. By subtracting the values measured for mock labelling conditions, the initial BRET change was corrected for labelling efficiency. Halo-corrected BRET changes were calculated by the division of the corrected and averaged values measured after ligand stimulation by the respective, corrected and averaged baseline values. Subsequently, this corrected BRET change was divided by the vehicle control for the final dynamic Δ net BRET change. These calculations were conducted using Excel 2016.

For the analysis of the respective BRET ratios before and after stimulation, the Halo-corrected and averaged BRET ratios before stimulation (baseline) and the Halo-corrected and averaged BRET ratios after stimulation (stimulated) are displayed as bar graphs. EC$_{50}$ values and the corresponding SEM for concentration-dependent β-arrestin recruitment were calculated with GraphPad Prism 7.03. using curves that were plotted from $n = 3$ independent experiments.

Receptors were stimulated as follows: human angiotensin II type 1 receptor (AT1R) with angiotensin II (AngII; Tocris 4474-91-3, in measuring buffer), human β2 adrenergic receptor (b2AR) with isoproterenol (Iso; Sigma-Aldrich I5627, in water) Epinephrine (Sigma-Aldrich E4642, in measuring buffer) and Norepinephrine (Sigma-Aldrich 74488, in measuring buffer), human b2AR with an exchanged C-terminus of the vasopressin type 2 receptor (b2V2) with Iso, human complement 5a receptor 1 (C5aR1) with C5aR-agonist (AnaSpec AS65121, in measuring buffer), human muscarinic 1–5 acetylcholine receptors (M1R, M2R, M3R, M4R and M5R) with acetylcholine (ACh; Sigma-Aldrich A6625, in measuring buffer), murine MOP with [D-Ala$^2$, N-MePhe$^4$, Gly-ol]-enkephalin (DAMGO; Tocris 1171, in water), human parathyroid hormone 1 receptor (PTH1R) with parathyroid hormone (1-34) (PTH(1-34); Bachem 4011474, in measuring buffer) and human vasopressin type 2 receptor (V2R) with [Arg$^8$]-vasopressin (AVP; Tocris 2935, in water). The transfected β-arrestins are of bovine origin. β-arrestin1 constructs lacking the finger loop region (dFLR) were generated as described in Cahill et al.[32] by site-directed mutagenesis. The corresponding β-arrestin2 constructs were designed homologously.

If not further elaborated, the utilised cDNAs were obtained from the cDNA resource centre (www.cDNA.org) or Addgene. The Halo-Tag or NanoLuc genes were acquired from Promega and were genetically fused to the respective N- or C-termini.

In the case of cmpd101 (Tocris 15777006, in DMSO) and pindolol (Sigma-Aldrich, P0778, in 0.1 M HCl) inhibitor experiments, the β-arrestin recruitment was induced with either 1 μM isoproterenol (in case of the b2AR) or 10 μM ACh (in case of the M2R) after a 10-min incubation period with different concentrations of cmpd101 or pindolol.

In the case of the Gö6983 (Tocris, 2285, in DMSO) experiments, the cells were pre-incubated with 500 nM of the inhibitor at 37 °C for 1 h and subsequently stimulated with different concentrations of AngII.

For the inverse agonist or antagonist experiments conducted for the V2R and AT1R, the cells were treated with the indicated concentration of Tolvaptan (1 μM) or Losartan (10 μM) 4 h after the transfer into 96-well plates and incubated overnight. The following washing steps, as well as the experimental procedure, were carried out with buffers containing the same concentration of the inverse agonist or antagonist, to guarantee unchanged receptor occupancy by the compounds. After the acquisition of baseline BRET ratios, the cells were stimulated with 1 μM AVP or 1 μM AngII, respectively. Control conditions without inverse agonist or antagonist treatment were measured again, side-by-side, to enable data comparability.

**Fluorometric assessment of GRK expression and functionality of GRK-YFP constructs**. For the assessment of GRK expression levels, C-terminal GRK-YFP fusions were constructed via isothermal plasmid assembly, keeping the identical

vector backbone. $1.6 \times 10^6$ ΔQ-GRK cells were seeded in 21 cm² dishes and transfected the next day with 1.5 μg of either GRK2-, 3-, 5- or 6-YFP fusion constructs. All transfections were conducted following the Effectene transfection reagent manual by Qiagen (#301427) and then incubated at 37 °C overnight. Into poly-D-lysine-coated 96-well plates (Brand, 781965), 40,000 cells were seeded per well. After 24 h of incubation at 37 °C, the fluorescence was assessed using a Synergy Neo2 plate reader (Biotek) and a corresponding YFP filter (excitation bandwidth 465–505 nm, emission 496–536 nm). To confirm the catalytic activity of the used GRK-YFP fusion constructs, GRK-specific β-arrestin2 recruitment intermolecular BRET assay featuring the PTH1R-Halo-Tag and β-arrestin2-NanoLuc was performed. In this case, cells were transfected as described in "intermolecular bioluminescence resonance energy transfer (BRET)" section. Instead of untagged GRK constructs, 0.25 μg of the GRK–YFP fusion constructs were transfected. After stimulation with PTH(1–34), the data were recorded and processed as described above.

**Statistical analysis of intermolecular BRET.** BRET ratios and fold changes are displayed as the mean of at least three independent experiments with error bars indicating the SEM. Statistical analysis was performed using Student's $t$-test or analysis of variance (ANOVA; one-way or two-way mixed model ANOVA), as well as appropriate multiple comparisons as indicated in corresponding figure legends. Data were prepared using Python 3.8.7 and statistical analysis was conducted in R 4.0.3[73]. A type I error probability of 0.05 was considered to be significant in all cases. Two-way mixed model ANOVA was performed using *ez* R package (Lawrence, MA. (2011) ez: Easy analysis and visualisation of factorial experiments. R package version 4.4-0. http://CRAN.R-project.org/package=ez) and multiple comparisons were conducted using the *multcomp* R package[74]. The clustering heatmap was generated using the *pheatmap* R package (Kolde, R. (2013). pheatmap: Pretty Heatmaps. R package version 1.0.12. http://CRAN.R-project.org/package=pheatmap.). Additionally, all code that was created for the statistical analysis of presented data can be accessed via 10.5281/zenodo.5764249.

**Intramolecular BRET.** ΔQ-GRK or Control cells were transfected with 1.2 μg untagged b2AR, 0.12 μg of β-arrestin2 FlAsH5-tagged biosensor C-terminally coupled to NanoLuc, 0.25 μg of either GRK2, 3, 5, 6 or empty vector as noted, following the Effectene transfection reagent protocol by Qiagen. 24 h after transfection, 40,000 cells were seeded per well into poly-D-lysine coated 96-well plates and incubated overnight at 37 °C. For this study, the FlAsH (fluorescein arsenical hairpin-binder)-labelling procedure previously described Hoffmann et al.[75] was adjusted for 96-well plates. In brief, the cells were washed twice with PBS, then incubated with 250 nM FlAsH in labelling buffer (150 mM NaCl, 10 mM HEPES, 25 mM KCl, 4 mM CaCl₂, 2 mM MgCl₂, 10 mM glucose; pH7.3), complemented with 12.5 μM 1,2-ethane dithiol (EDT) for sixty minutes at 37 °C. After aspiration of the FlAsH labelling or mock labelling solutions, the cells were incubated for 10 min at 37 °C with 100 μl 250 μM EDT in labelling buffer per well. In addition to the NanoLuc substrate, measurement and analysis of the BRET change was performed as described above (see Section "Intermolecular bioluminescence resonance energy transfer (BRET)").

**Microscopy.** The morphology of the generated cell clones was documented during regular cell culture procedures using phase-contrast microscopy at the Invitrogen EVOS FL Auto in 10× magnification.

Receptor internalisation of fixed cells stably expressing the MOP was analysed using confocal microscopy. ΔGRK2/3, ΔGRK5/6, ΔQ-GRK and Control cells were grown on poly-L-lysine-coated coverslips for 2–3 days. After the treatment with 10 μM DAMGO at 37 °C for 30 min, cells were fixed with 4% paraformaldehyde and 0.2% picric acid in phosphate buffer (pH 6.9) for 30 min at room temperature. Then coverslips were washed several times with PBS w/o Ca²⁺/Mg²⁺ buffer. After washing with 50% and 100% methanol for 3 min, cells were permeabilised with phosphate buffer for 2 h and then incubated with anti-HA antibody (7TM000HA, 7TM Antibodies (Jena, Germany), 1:500) followed by Alexa488-conjugated secondary antibody (1:2000) (Life Technologies, Thermo Fisher Scientific A11008). Specimens were mounted with Roti®-Mount FluorCare DAPI (Carl Roth, HP20.1) and examined using a Zeiss LSM510 META laser scanning confocal microscope.

Live cell experiments were performed to record the translocation of PTH1R, M5R, V2R or AT1R, β-arrestin and the early endosome marker Rab5 upon agonist stimulation at a Leica SP8 laser scanning confocal microscope operated with the Leica Application Suite X (version 3.5.5.19976). Therefore ΔGRK2/3, ΔGRK5/6, ΔQ-GRK and Control cells were transfected with 1 μg of the C-terminally CFP-fused receptor (in the case of AT1R, we included an HSP-export tag[76] and transfected 1.25 μg), 0.5 μg of β-arrestin-YFP, 0.5 μg of Rab5-mCherry (kindly provided by Tom Kirchhausen (Harvard Medical School, Boston, USA)) and 0.25 μg GRK expression constructs (as indicated) in a 21 cm² dish, according to the Effectene transfection reagent manual by Qiagen. After 24 h, 700,000 transfected cells were seeded onto poly-D-lysine-coated glass coverslips in 6-well plates. Another 24 h later, the coverslips were washed twice with measuring buffer and subsequently imaged before and after stimulation for 15 min with either 100 nM PTH(1-34), 100 μM ACh, 100 nM AVP or 100 nM AngII as indicated. CFP was excited at a wavelength of 442 nm, YFP at 514 nm and mCherry at 561 nm. The images were acquired with a 63× water immersion objective, with zoom factor 3, line average 3 and 400 Hz in 1024 × 1024

pixel format. Subcellular features of the acquired images were segmented and quantified using an ImageJ-based software (ImageJ version 1.52p) called segmentation and quantification of subcellular shapes (Squassh). Utilising Squassh's deconvolution, denoising and segmentation of the three fluorescence channels present in each image, the raw data readout was then eligible for analysis using the R-based software SquasshAnalyst as described by A. Rizk[77,78]. All image-derived data in this study were processed and analysed with this method and are presented as fold change in co-localisation signal. Statistical analysis of quantified microscopy data was performed using GraphPad Prism 7.03. Unstimulated and stimulated co-localisation was compared using paired t-test. To identify significantly increased co-localisation under unstimulated conditions, quantified co-localisation was compared using ANOVA and two-sided Dunnett's test.

**Enzyme-linked immunosorbent assay (ELISA).** Receptor internalisation was quantified using a linear surface receptor ELISA that has been characterised extensively[79,80]. Equal numbers of ΔGRK2/3, ΔGRK5/6, ΔQ-GRK and Control cells stably expressing HA-tagged murine MOP were seeded onto poly-L-lysine-coated 24-well plates for 2-3 days. Then, cells were pre-incubated with anti-HA antibody (7TM000HA), obtained from 7TM Antibodies (Jena, Germany, 1:500) for 2 h at 4 °C. After 30 min treatment with 10 μM DAMGO at 37 °C, the cells were fixed with 4% paraformaldehyde and 0.2% picric acid in phosphate buffer (pH 6.9) for 30 min at room temperature and incubated with peroxidase-conjugated anti-rabbit antibody (Cell Signalling technology #7074, 1:1500) overnight at 4 °C. After washing, the plates were developed with ABTS solution (Sigma-Aldrich A3219) and analysed at 405 nm using a microplate reader.

**Label-free DMR biosensing.** Dynamic mass redistribution (DMR) experiments were performed using the Corning Epic (Corning, NY, USA) biosensor technology as previously described in detail[38,81–86]. In short, for DMR detection $9 \times 10^5$ ΔQ-GRK cells were seeded into 21 cm² dishes and cultured until reaching a confluence of 60–80%, which is critical for the maintenance of a consistent proliferation phenotype and for comparable transfection efficiencies. Subconfluent cells were then transiently transfected with empty vector (pcDNA3) or expression plasmids (pcDNA3-based) encoding for the AT1R (HSP-AT1R-CFP), β-arrestin2, GRK2 or GRK6. The next day, transfected cells were seeded at a density of $2 \times 10^4$ cells per well into Corning Epic biosensor microplates and incubated overnight (37 °C, 5% CO₂). Prior to DMR detection, cells were washed three times with HBSS buffer containing 20 mM HEPES and 0.1% fatty acid-free bovine serum albumin (Sigma-Aldrich). Cells were then placed into the Epic-DMR-reader, operated with the software Epic Imager 2012 and equilibrated for 1 h at 37 °C to achieve baseline stabilisation. Compounds diluted in the same buffer were added to the biosensor plate using the CyBio Selma semi-automatic pipettor (Analytik Jena AG) and ligand-induced DMR alterations were monitored as picometre (pm) wavelength shifts for at least 3600 s in 15 s intervals. Real-time DMR recordings are means + SEM of three technical replicates and were corrected by the pm wavelength shifts obtained in empty vector transfectants. Concentration-effect curves are means ± SEM of $n = 3$ independent biological replicates and were derived from the area under the curve between zero and 1800 s using a three-parameter logistic equation and the GraphPad Prism (8.4.3) software. Curves were fitted with "bottom" constrained to zero while all other settings were left to their default values. Two way ANOVA was used for statistical analysis.

**Cell line availability.** All created cell lines will be made available upon request.

**Reporting summary.** Further information on research design is available in the Nature Research Reporting Summary linked to this article.

## Data availability

All data supporting the findings of this study are available within the article and its supplementary information files. Additional information, relevant data and unique biological materials will be available from the corresponding author upon reasonable request. Source data are provided with this paper.

## Code availability

Custom code was created for general statistical analysis as well as for the analyses presented in Fig. 3i and Fig. 7. The code is available via Github and under https://doi.org/10.5281/zenodo.5764248.

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

## Acknowledgements

We want to thank Ulrike Schiemenz and Nina Kathleen Blum for assistance with the MOP-internalisation studies, Dr. Aurélien Rizk for the help in microscopy data analysis, Prof. Tom Kirchhausen for providing the Rab5-mCherry plasmid, and the Core Facility Flow cytometry of the FLI—Leibniz Institute for Age Research, Jena, for sorting of the stable cell lines.

## Author contributions

J.D. engineered all GRK knockout cells; J.D., R.S.H. and C.H. developed the concept and designed the experiments; J.D., R.S.H., E.S.F.M., M.R., S.B, C.Z., L.K., J.F. and V.W. conducted the experimental work; J.D., R.S.H., E.S.F.M. and M.R. compiled the data; J.Z. designed, conducted and analysed AT1R targeted DMR measurements, supervised by E.K.; S.F. and A.K. planned, conducted and analysed the MOP internalisation experiments with support from E.M.-T.; C.H. supervised the project; S.S. provided phosphosite specific antibodies; J.D., R.S.H., E.S.F.M., M.R. and C.H. wrote the paper; all other authors critically revised the paper and gave final approval.

## Funding

C.H. was supported by the European Regional Development Fund (Grant ID: EFRE HSB 2018 0019), the federal state of Thuringia, the Deutsche Forschungsgemeinschaft (Grants: CRC166, *ReceptorLight*, project C02 and *Polytarget*; SFB1278: 316213987, project D02). J.D. is additionally funded by the University Hospital Jena IZKF (Grant ID: MSP10). E.K. gratefully acknowledges the support of this work by the DFG-funded Research Unit FOR2372 with the grants KO 1582/10-1 and KO 1582/10-2. Open Access funding enabled and organized by Projekt DEAL.

## Competing interests

S.S. is the founder and scientific advisor of 7TM Antibodies GmbH, Jena, Germany. The remaining authors declare no competing interests.
