## [Peer Review File · Nature Communications]

GPCR kinase knockout cells reveal the impact of individual GRKs on arrestin binding and GPCR regulationREVIEWER COMMENTS

Reviewer #1 (Remarks to the Author):

The study clearly consists of two parts. One, creation of a panel of HEK293 cells lacking individual GRKs and their various combinations, including cells lacking GRK2/3/5/6. This part is valuable for the whole GPCR field and constitutes a tour-de-force. However, in and of itself it is better suited for Nature Methods. Two, the use of these cells +/- overexpression of individual GRKs and PKC inhibition to determine relative importance of individual GRK subtypes in b-arrestin1/2 binding to a wide panel of GPCRs belonging to classes A and B that couple to different G proteins. The results of these extensive meticulously performed studies are a lot less definitive than one would wish. The authors do present a tentative classification of GPCRs by GRK regulation, while finding that both vasopressin and angiotensin II receptors fall into a class of their own due to agonist-independent arrestin recruitment. It is quite likely that the results are not clear-cut due to the nature of things, and the authors would be well advised to present their data from this perspective, rather than attempt to make the results look clearer than they really are. With this change of presentation angle the data would be of great interest to the GPCR field. Some less global revisions would also improve the manuscript.

Experimental

1. The authors should acknowledge that their GRK-deficient cells might express different isoforms of GRK4, or present data showing that GRK4 is not expressed in HEK293 cells. Also, why no isoform of GRK4 was tested?

Presentation

2. The authors should compare their analysis of patterns of potentially phosphorylatable residues in GPCRs with recently reported "phosphorylation codes" needed for arrestin binding (Cell. 2017 Jul 27;170(3):457-469.e13). Looks like the data suggest that those phosphorylation codes do not constitute the whole story. The authors should state this explicitly.

3. Two systems of arrestin names are used. B-arrestin1 and 2 are also called arrestin-2 and -3, respectively. The authors should provide a translation.

4. Commonly used abbreviation for b2-adrenergic receptor is b2AR, not β 2ADR.

5. The authors repeatedly demonstrate that b-arrestin2 is less dependent on GPCR phosphorylation than b-arrestin1. Greater promiscuity of b-arrestin2 was noted before and its structural basis was demonstrated experimentally (J Mol Biol. 2011 Feb 25;406(3):467-78) and by modeling (ACS Chem Neurosci. 2016 Sep 21;7(9):1212-24). This should be mentioned.

6. Some editing is needed. Abstract and title, "arrestin-binding" should be "arrestin binding"; line 198, "when" should be "in cells"; line 342, "disclosure" should be "discovery"; line 517, "confinements" should be "locations" or "microdomains"; etc.

Reviewer #2 (Remarks to the Author):

Drube and colleagues have generated a comprehensive panel of single, double and quadruple GRK2/3/5/6 knockout HEK293 cells, and utilized these to assess a panel of G protein coupled receptors to categorize GRK-dependent beta-arrestins interactions with receptor and signaling. They also compare quad-knockout cells re-expressing individual GRKs at a constant level. Using these tools, they examine the GRK-dependence of phospho-sites in the mu opiate receptor in detail, and the beta-arrestin 1 and 2-recruitment by a panel of receptors. The authors conclude that GPCRs sort into groups that are predominantly GRK2 or 3 regulated, or are regulated about equally by any of the 4 ubiquitous GRKs; and that individual GRKs induce differential beta-arrestin association with receptors (tail vs core). This is an interesting manuscript that describes a useful set of tools for understanding GPCR regulation and signaling bias, and details some applications. The field will find this a useful addition.

Comments

1. The interpretations of specificity rely on knowing the expression level of the GRKs in these assays

(either endogenous or when overexpressed). Endogenous GRK expression levels for HEK cells are reported using an independent dataset using someone else's HEK cells (Fig 1d), not for these cells in this lab. Since HEK293 cell isolates can vary considerably in their expression of multiple GPCR signaling components (see for example PMID), the authors should directly measure the GRK levels in their particular cell lines. One way to do this is comparing western blots on their cells versus a standard curve of known quantities of each GRK (recombinant proteins are available commercially).

2. Overexpressed GRK levels are compared using a tagged version of each GRK, and the authors demonstrate that they are comparing comparable levels of re-expression (Fig 1f). However, there is no information on just how high these overexpressed levels might be – and this is a concern as data interpreted as a lack of specificity (e.g., no GRK5/6-only-dependent events, line 258) may instead be a result of massive overexpression compared to endogenous GRK. The interpretation of much of Fig 3 depends on the re-expressed GRKs not being overwhelming. The authors need to measure the overexpression levels of each GRK relative to its endogenous level. This can be done by western blotting using multiple dilutions of the overexpression lysate until the signal is equal to that of the endogenous GRK lysate (fold overexpression).

3. Statistics appear to be only selectively applied to the data, with many figures not showing statistics and the text descriptions making suggestive but not definitive statements (“apparent increase” on line 189, etc). This is particularly true of dose-response data purporting to show EC₅₀ shifts, but also phospho-site relative phosphorylation and receptor internalization data. The statistical analysis needs to be more consistent and rigorous.

4. The concerns about interpreting data from GRK overexpression is also a major factor in understanding the data in Fig 3 with AngII receptor recruitment of beta-arrestins. Specifically, are the effects on basal and stimulated recruitment by GRKs 5/6 seen with endogenous levels of the kinases, or only when they are overexpressed? That the baseline level of recruitment changes so much raises a serious concern that the basal activity of overexpressed GRKs is preventing the ability to discern stimulated recruitment.

5. Minor, but the plot for Supp Fig 1e has out of order residue locations. The authors appear to be separating S263 from the others as a non-GRK site by putting it last, but this is confusing to anyone who looks at the figure before the text (as many do), and it would be equally separate if plotted first, as it would be if sorted by receptor residue number.

Reviewer #3 (Remarks to the Author):

In this study Drube and co-authors have developed a platform to investigate the role of individual GRKs in GPCR-arrestin interactions. They have used a systematic CRISPR/Cas9 approach to generate an impressive collection of cell lines that included single, double, triple and quadruple KOs of the most ubiquitously expressed GRKs. They have then combined the use of these cells with a BRET system to assess GPCR-arrestin interactions to investigate i) the role of endogenous GRKs in promoting such interactions and ii) the effect of individual GRK2 overexpression in GPCR regulation. In doing these experiments on a group of twelve GPCRs they have encountered several interesting findings, and, perhaps most importantly, they have shed light into the determinants (or lack of thereof) guiding receptor phosphorylation and its subsequent intracellular consequences. This study has the potential to extend beyond the “phosphorylation barcode” hypothesis for GPCRs. Although the manuscript has a huge amount of data, the authors make an effort to present it clearly and the balance of data between main figures and supplementary is adequate.

Comments:

1- In line 147, the authors state that “data which could be described by a curve-fit will be further interpreted as functional recruitment.” This seems to be a weak criterion to classify responses and is particularly apparent in some panels in Supp Figs 3 and supported by the bar graphs in supp fig4.

GraphPad Prism will sometimes fit curves over flat lines; thus, the CI of the individual curve fits, or statistical analysis on maximal responses compared to vehicles in the instances where the authors see a “response” should be provided to provide confidence on such observations.

2- In line 190 the authors discuss the relative efficacy of the endogenous ligands norepinephrine and epinephrine. The data in supp fig 2 seems to show that these two agonists are partial relative to Iso. This is in contrast to reference 27, where epinephrine is a full agonist. It is noted that the experimental set up in ref 27 and in the current manuscript is very different and perhaps the source of this discrepancy? In line with this, the authors should comment on the choice of assay to monitor GPCR-arrestin interaction there are multiple formats to detect such interactions (BRET, NanoBiT, FRET, DiscoverX, Tango...) and assay format may impact data interpretation.

3- If I understand correctly, data plotted in the concentration response curves and bar graphs in Fig 3a and b respectively, should be related. Roughly the ratio stimulated/baseline in the bar graph should approx give the saturation asymptote of the CRC. This doesn't seem to be the case for DeltaQ-GRK +EV V2R barr1. According to the protocols the difference between these graphs is the additional step of vehicle normalisation in the curve that is not present in the bar graph. Can the authors clarify? Is the vehicle effect then very different in that specific condition?

4- The mechanism proposed by the authors whereby individual kinases facilitate “constitutive” engagement of arrestins to V2R and AT1Rs is very interesting. How do the authors hypothesise the kinases mediate receptor phosphorylation that regulates arrestin recruitment? Can it be constitutive activity of the receptor? Perhaps this would explain the b2V2 chimera result, whereby no constitutive activity is present from b2 as opposed from V2 and hence no basal phosphorylation? Does V2R show constitutive activity? If so, it would be interesting to test whether an inverse agonist prevents the raise in baseline observed upon overexpression of individual GRKs. Alternatively, or in parallel, use of kinase dead GRKs for V2R as with the AT1R would shed some insight into this question.

5- Line 321 - “We observed the most prominent difference between angiotensin II (AngII)-induced β -arrestin1 and 2 recruitment to the AT1R, in Δ Q-GRK, as this condition features a pronounced higher recruitment of β -arrestin2 (Supplementary Figure 3g)” – this statement is better supported by the data in Supp Fig 4.

6- The paragraph describing GRK2/3 and PKC effects on AT1R-arrestin interactions (starting line 350) needs to make it clearer that is its specifically referring to ligand-induced interactions. While the authors are very familiar to the terminology they use, the reader may need a bit more guidance.

7- Line 414 “While the V2R and AT1R are both regulated by all tested GRKs, they exhibited substantial β -arrestin pre-coupling and therefore constitute a subgroup of GRK2/3/5/6-regulated receptors”. This statement should clarify that the pre-coupling was observed upon GRK overexpression.

8- A considerable section of the discussion attempts to find correlations between phosphorylation motifs and GRK specificity. A recent attempt to identify different functional classes of phosphorylation sites suggested the existence of “key”, “inhibitory” and “modulatory” phosphorylation sites for arrestin binding (Mayer et al Nat Comms 2019). Similarly, computational and biophysical studies on GPCR phosphorylation patterns have recently addressed the question of “phosphorylation barcodes” (e.g. Latorraca et al., Cell 2020). Some discussion around the context of this previous work and the current results should be included.

9- Minor comment: Line 494 – “whereas” should be “whereby”?

For convenience of the reviewers, we provide a list of changes in display items.

List of changes in display items

- Figure 1:**
- d** RNA sequencing data of the human protein atlas has been replaced with relative GRK protein expression in Control cells used in this study, determined by western blot.
 - h, i, j** EC₅₀ changes were statistically analysed and significant changes were indicated using asterisks as described in the figure legend.
 - f** To confirm similar expression levels of GRK-YFP constructs, the differences in measured YFP fluorescence were statistically analysed and found to be not significant.
- Figure 2:** No changes.
- Figure 3:** No changes.
- Figure 4:** No changes.
- Figure 5:** No changes.
- Supplementary Figure 1:**
- b, c** Growth rates of the knock out cell lines and relative GRK expression in these cell lines were statistically analysed and significant changes were indicated using asterisks as described in the figure legend.
 - e** The relative phosphorylation data for the MOP are now presented in order of the amino acid position, with pS363 being the first bar for each analysed cell line. In addition, differences in relative phosphorylation of the specific sites were statistically analysed and significant changes were indicated using asterisks as described in the figure legend.
 - g** Differences in MOP internalisation between the analysed cell lines compared to the Control cells were statistically tested and significant changes were indicated using asterisks as described in the figure legend.
- Supplementary Figure 2:** **b, c** A representative western blot of the overexpression of GRK2, 3, 5, and 6 in Δ Q-GRK cells in comparison to endogenous levels and a relative quantification presented as fold overexpression of respective signal from endogenously expressed GRKs were included.
- Supplementary Figure 3:** Data was previously presented in **Supplementary Figure 2**.
- Supplementary Figure 4:** Data was previously presented in **Supplementary Figure 3**.
- Supplementary Figure 5:** Data was previously presented in **Supplementary Figure 4**.
- Supplementary Figure 6:** Data was previously presented in **Supplementary Figure 5**.
- Supplementary Figure 7:** Data was previously presented in **Supplementary Figure 6**.
- Supplementary Figure 8:** New experimental data of β -arrestin1 recruitment to the V2R using kinase dead (KD) variants of GRK2 or 6 were included.

- Supplementary Figure 9:** Data was previously presented in **Supplementary Figure 7**.
e,g The data presented in these panels were statistically analysed to compare the level of basal interaction of the β -arrestin2 with the AT1R under the indicated conditions. Significantly elevated basal interaction compared to the Δ Q-GRK baseline is pointed out by asterisks as described in the figure legend.
- Supplementary Figure 10:** Data was previously presented in **Supplementary Figure 8**.
- Supplementary Figure 11:** Data was previously presented in **Supplementary Figure 9**.
- Supplementary Figure 12:** New experimental data of β -arrestin2 recruitment to the V2R and AT1R after incubation over night with an inverse agonist or antagonist (Tolvaptan or Losartan, respectively) were now included.
- Supplementary Figure 13:** Data was previously presented in **Supplementary Figure 10**.
- Supplementary Figure 14:** New experimental data of β -arrestin2 recruitment to phosphorylation deficient variants of the AT1R (S347A, S348A, S347A/S348A) were now included. The mutated positions were proposed to be inhibitory phosphorylation sites of the AT1R by Mayer *et al*. The β -arrestin2 recruitment data to the wild type AT1R is shown again to allow a direct comparison.
- Supplementary Table 1:** All BRET data presented in this study was statistically analysed to compare the respective highest ligand stimulation with vehicle-stimulated values. Only those data sets, which show a significant increase in this comparison are further interpreted as functional β -arrestin recruitment. The displayed concentration-response curves have been adjusted throughout the manuscript, according to the data in this table.
- Supplementary Table 2:** This table has been previously presented as **Supplementary Table 1**.
- Supplementary Table 3:** This table has been previously presented as **Supplementary Table 2**.
- Supplementary Table 4:** This table has been previously presented as **Supplementary Table 3**.

Point-to-point reply to the reviewers' comments:

We thank the three reviewers for their constructive and fair feedback regarding the manuscript at hand. In the revised version of Drube et al., we comprehensively addressed the reviewer's criticism and included several new experiments to strengthen our line of reasoning. We sincerely believe that the well-balanced suggestions made by the referees improved our existing work.

Reviewer #1 (Remarks to the Author):

The study clearly consists of two parts. One, creation of a panel of HEK293 cells lacking individual GRKs and their various combinations, including cells lacking GRK2/3/5/6. This part is valuable for the whole GPCR field and constitutes a tour-de-force. However, in and of itself it is better suited for Nature Methods. Two, the use of these cells +/- overexpression of individual GRKs and PKC inhibition to determine relative importance of individual GRK subtypes in β -arrestin1/2 binding to a wide panel of GPCRs belonging to classes A and B that couple to different G proteins. The results of these extensive meticulously performed studies are a lot less definitive than one would wish. The authors do present a tentative classification of GPCRs by GRK regulation, while finding that both vasopressin and angiotensin II receptors fall into a class of their own due to agonist-independent arrestin recruitment. It is quite likely that the results are not clear-cut due to the nature of things, and the authors would be well advised to present their data from this perspective, rather than attempt to make the results look clearer than they really are. With this change of presentation angle the data would be of great interest to the GPCR field. Some less global revisions would also improve the manuscript.

*We thank the referee for his or her detailed synopsis of the manuscript at hand, which accurately covers our most striking findings. Additionally, we appreciate that our presented analyses and the developed tools were found to be of value for the scientific GPCR community. We agree that for some receptors, our assessment did not yield clear-cut results and we are grateful as the reviewer also acknowledged the difficulties of interpreting and presenting complex findings to create a compelling and coherent study, while abstaining from over interpretation of these biological data. To address this global point of criticism, we revised several paragraphs in the **results (lines 170-172)** and **discussion (lines 545-547, 551-556)** sections to more accurately describe our initially presented data. Additionally, we want to direct the reviewer's attention towards a previously existing discussion paragraph (now **lines 524-528**). We disagree that we attempt to present the data for the V2R and AT1R as clear-cut. We have specifically addressed these receptors in great detail in **Figure 3 and 4** as well as in **Supplementary Figure 7, 8, 9, 10, 11, 12, and 14**. We hope that with these changes, our study will be justifiably received in the proper context of scientific literature.*

Furthermore, we addressed all specific points of critique provided by the reviewer on the following pages.

Experimental

1. The authors should acknowledge that their GRK-deficient cells might express different isoforms of GRK4, or present data showing that GRK4 is not expressed in HEK293 cells. Also, why no isoform of GRK4 was tested?

*We thank the reviewer for pointing out the interesting and highly relevant topic of GRK4 contributions to GPCR regulation. Indeed, it is possible that HEK293 cells might express GRK4 and we agree that the submitted manuscript fails to acknowledge this. To address this shortcoming, we clarified the putative functionality and impact of GRK4 in the **introduction** section (**lines 67-76**). Moreover, we indeed tested different GRK4 constructs for their ability to facilitate β -arrestin recruitment. With the graph shown below, we want to present the reviewer with some initial results that are not destined for imminent publication, but might explain our reasoning of why we excluded GRK4 from our study.*

PTH1R + β arr2

As a model receptor that features very stable β -arrestin recruitment and co-localisation, facilitated by GRK2, 3, 5, and 6, we initially chose the PTH1R as a suitable GPCR to test the functionality of GRK4. Interestingly, our data suggest that the overexpression of GRK4, unlike all other tested GRK isoforms (**Supplementary Figure 4f**), is not able to promote increased β -arrestin recruitment at the PTH1R. This is especially emphasised as the overexpression of the GRK4 kinase dead (KD) variant yields virtually identical data. These experiments, combined with the limited expression profile of GRK4 throughout human physiology, convinced us to focus our current study on GRK2, 3, 5, and 6. Nevertheless, we are working on follow-up studies that elaborate on the functionality of GRK4 in developmental biology and specifically organogenesis, as we agree that more research has to be done to clarify the role of GRK4 on the cellular and organism scale.

2. Presentation: The authors should compare their analysis of patterns of potentially phosphorylatable residues in GPCRs with recently reported “phosphorylation codes” needed for arrestin binding (Cell. 2017 Jul 27;170(3):457-469.e13). Looks like the data suggest that those phosphorylation codes do not constitute the whole story. The authors should state this explicitly.

We thank the referee for bringing this shortcoming to our attention. We apparently failed to cite this publication in our initial submission, which is especially bothersome as we were actually using this information as a primary source for our evaluation of phosphorylation codes. This shortcoming has been corrected and Zhou et al. 2017 is now aptly cited and referred to. Additionally, we agree with the reviewer that there are more determinants influencing arrestin interactions than the phosphorylation state of a GPCR. The paragraphs in the **discussion (line 530)** section have been changed to more clearly describe this finding.

3. Two systems of arrestin names are used. B-arrestin1 and 2 are also called arrestin-2 and -3, respectively. The authors should provide a translation.

*We thank the reviewer for his or her insightful comment. To address this point, we added a clarification of arrestin nomenclature in the **introduction** section (lines 67-76).*

4. Commonly used abbreviation for b2-adrenergic receptor is b2AR, not β 2ADR.

To address the referee's suggestion, we gladly changed the acronym used for the β 2-adrenergic receptor to "b2AR" instead of " β 2ADR".

5. The authors repeatedly demonstrate that b-arrestin2 is less dependent on GPCR phosphorylation than b-arrestin1. Greater promiscuity of b-arrestin2 was noted before and its structural basis was demonstrated experimentally (J Mol Biol. 2011 Feb 25;406(3):467-78) and by modeling (ACS Chem Neurosci. 2016 Sep 21;7(9):1212-24). This should be mentioned.

*We thank the referee for pointing out these highly relevant and interesting publications. To address this comment, we included the two publications in the **discussion** (lines 434-450) section and connected these intriguing observations with our novel findings. We agree with the reviewer that these previously published results are not only in agreement with our findings, but actually deliver an attractive structural explanation.*

6. Some editing is needed. Abstract and title, "arrestin-binding" should be "arrestin binding"; line 198, "when" should be "in cells"; line 342, "disclosure" should be "discovery"; line 517, "confinements" should be "locations" or "microdomains"; etc.

We again thank the reviewer for making us aware of these semantic shortcomings of our manuscript text. The suggested changes have been implemented accordingly in the revised version of Drube et al.

Reviewer #2 (Remarks to the Author):

Drube and colleagues have generated a comprehensive panel of single, double and quadruple GRK2/3/5/6 knockout HEK293 cells, and utilized these to assess a panel of G protein coupled receptors to categorize GRK-dependent beta-arrestins interactions with receptor and signaling. They also compare quad-knockout cells re-expressing individual GRKs at a constant level. Using these tools, they examine the GRK-dependence of phospho-sites in the mu opiate receptor in detail, and the beta-arrestin 1 and 2-recruitment by a panel of receptors. The authors conclude that GPCRs sort into groups that are predominantly GRK2 or 3 regulated, or are regulated about equally by any of the 4 ubiquitous GRKs; and that individual GRKs induce differential beta-arrestin association with receptors (tail vs core). This is an interesting manuscript that describes a useful set of tools for understanding GPCR regulation and signaling bias, and details some applications. The field will find this a useful addition.

We thank the reviewer for his or her comprehensive summary of our main findings. Additionally, we are grateful for the overall positive evaluation of our work. In accordance with the reviewer's constructive comments, we have incorporated the suggested experiments and changes to the manuscript as detailed below.

Comments

1. The interpretations of specificity rely on knowing the expression level of the GRKs in these assays (either endogenous or when overexpressed). Endogenous GRK expression levels for HEK cells are reported using an independent dataset using someone else's HEK cells (Fig 1d), not for these cells in this lab. Since HEK293 cell isolates can vary considerably in their expression of multiple GPCR signaling components (see for example PMID), the authors should directly measure the GRK levels in their particular cell lines. One way to do this to comparing western blots on their cells versus a standard curve of known quantities of each GRK (recombinant proteins are available commercially).

*We unconditionally agree with the referee's comment and established a Western blot-based system that allows the relative quantification of endogenously expressed GRK2, 3, 5, and 6. Instead of using purified commercially available protein to provide a correlate for relative quantification, we genetically attached HA-tags to our GRK expression constructs and prepared lysates of transfected ΔQ-GRK cells. Utilising the comparable signal retrieved from probing the different lysates with an anti-HA antibody, we furthermore prepared dilutions with equimolar levels of HA-tagged GRKs. These standardised dilutions were then run side-by-side with lysates prepared from untransfected Control cells and probed with the respective GRK-isoform specific antibodies. The quantified signal of the endogenously expressed individual GRKs was then divided by the individual standard – allowing the direct comparison of the GRK expression levels among each other. These data were then normalised to GRK2 expression levels and are now plotted in **Figure 1d**, replacing the initially presented transcriptomic data.*

*Indeed, as the reviewer aptly pointed out, our utilised HEK293 Control cells exhibit a different GRK protein expression pattern as initially suggested by the mRNA quantification. Since the previously used dataset underestimated the found GRK2 expression level, we changed parts of the **results (lines 165-168)** section and emphasised that individual GRK isoforms display differential affinities for specific GPCRs.*

*A lot of work was allocated to solve this problem, as this method of quantification requires the establishment of a robust Western blot setup and – most importantly – reliable antibodies to detect the different GRK subtypes. In our view, the optimisation of GRK isoform detection and the presented quantification technique represent valuable resources for the GPCR community. Hence, we prepared a separate manuscript (Reichel et al., 2021), added as reference 26, which describes the workflow in detail. The revised version of Drube et al. now refers directly to the mentioned pre-print in order to maintain a manageably sized **methods** section (**line 1041**).*

2. Overexpressed GRK levels are compared using a tagged version of each GRK, and the authors demonstrate that they are comparing comparable levels of re-expression (Fig 1f). However, there is no information on just how high these overexpressed levels might be – and this is a concern as data interpreted as a lack of specificity (e.g., no GRK5/6-only-dependent events, line 258) may instead be a result of massive overexpression compared to endogenous GRK. The interpretation of much of Fig 3 depends on the re-expressed GRKs not being overwhelming. The authors need to measure the overexpression levels of each GRK relative to its endogenous level. This can be done by western blotting using multiple dilutions of the overexpression lysate until the signal is equal to that of the endogenous GRK lysate (fold overexpression).

*Analogous to the previous comment, we thoroughly agree with the reviewer's opinion and provided a relative quantification of the degree of GRK overexpression (**Supplementary Figure 2**). The presented graph describes the fold increase in GRK protein levels between endogenous and overexpressed conditions, for each GRK isoform respectively. Our data suggest that GRK2 and 6 exhibit the lowest fold increase of overexpression over endogenous levels, while showing the highest abundance of endogenously expressed protein (**Figure 1d**). We hope that the reviewer appreciates that these data ultimately indicate a comparable overexpression of all four GRK isoforms, as additionally supported by the fluorometrical assessment of GRK-YFP fusion constructs in **Figure 1f**. Furthermore, we acknowledge the referee's concerns regarding the interpretation of β -arrestin recruitment experiments conducted in the presence of overexpressed GRKs. These points are comprehensively addressed in the related comment 4.*

3. Statistics appear to be only selectively applied to the data, with many figures not showing statistics and the text descriptions making suggestive but not definitive statements ("apparent increase" on line 189, etc). This is particularly true of dose-response data purporting to show EC50 shifts, but also phospho-site relative phosphorylation and receptor internalization data. The statistical analysis needs to more consistent and rigorous.

*We thank the referee for pointing out these shortcomings of our statistical analysis. In our initial submission, we focused on the comprehensive statistical evaluation of our main data set that comprises β -arrestin recruitment measurements. Nevertheless, we agree with the reviewer that especially the mentioned datasets were lacking statistics. Hence, we included the appropriate statistical tests for the suggested datasets as stars indicated in the respective figures: **Figure 1h-j**, **Supplementary Figure 1e** and all other relevant results: **Figure 1f**; **Supplementary Figure 1b, c, g**; **Supplementary Figure 9e, g**. Furthermore, in the revised manuscript we only interpreted BRET data as functional β -arrestin recruitment if there is a significant difference in BRET after ligand application compared to vehicle. Thus, concentration-response curves were only fitted for these conditions. Results of this statistical analysis are presented in **Supplementary Table 1**. Additionally, appropriate statistical tests were also conducted for datasets which were added during the revision process: **Figure 1d**, **Supplementary Figure 2c**, **Supplementary Figure 8**, **Supplementary Figure 12** and **Supplementary Figure 14**.*

4. The concerns about interpreting data from GRK overexpression is also a major factor in understanding the data in Fig 3 with AngII receptor recruitment of beta-arrestins. Specifically, are the effects on basal and stimulated recruitment by GRKs 5/6 seen with endogenous levels of the kinases, or only when they are overexpressed? That the baseline level of recruitment changes so much raises a serious concern that the basal activity of overexpressed GRKs is preventing the ability to discern stimulated recruitment.

We thank the reviewer for raising this interesting and very valid concern. Here, we would like to take this opportunity to clarify our argumentative reasoning. Experiments using the overexpression of GRKs have been exclusively conducted using our Δ Q-GRK cell line. For these experiments, the aim was to specifically rescue the knockout of a single GRK isoform and thus, we used a minimal amount of GRK plasmid DNA for our transfections (only 1/10 of the total transfected DNA). But, as the reviewer aptly

pointed out, even this conservative use of plasmid DNA increases GRK expression to higher levels as seen for their endogenous expression (**Supplementary Figure 2**). Still, we would like to argue that the overexpression of these kinases does not limit their molecular capabilities to phosphorylate their respective targets, as illustrated by our b2AR and PTH1R GRK-specific β -arrestin recruitment experiments. Both receptors exhibit increased β -arrestin recruitment upon overexpression of GRK2, 3, 5, and, 6. Interestingly, the same expression levels of recombinant GRKs have a different influence for other tested receptors. Especially GRK5 and 6 seem to prefer Class B GPCRs, whereas the overexpression of GRK2 or 3 increases β -arrestin recruitment for most receptors. For the AT1R, the overexpression of GRK5 or 6 yields significantly higher baseline BRET ratios, as compared to the values measured in Δ Q-GRK+EV. As pointed out by the referee, this constitutive interaction can only be mildly increased upon agonist application.

This is not the case for the endogenous “leftover” expression of either GRK5 or 6, as demonstrated by utilisation of Δ GRK2,3,6 or Δ GRK2,3,5 cell lines (as seen in **Supplementary Figure 9g**) and only becomes apparent if compared to the appropriate control condition (Δ Q-GRK+EV). After we conducted multiple control experiments, featuring kinase-dead (KD) conditions, confocal microscopy and label-free dynamic mass redistribution (DMR), we found that the overexpression of GRK5 or 6 does not only lead to ligand-independent β -arrestin complex formation (mostly in a “hanging” configuration, as the AT1R is presumably still in an inactive, yet phosphorylated conformation), but also to constitutive receptor internalisation and intracellular retention. Interestingly, this does not seem to be an exclusive feature of GRK5 and 6, but mostly depends on the receptor itself, as we found that the V2R behaves very similarly, but shows this response upon overexpression of any of the tested GRKs (GRK2, 3, 5, or 6). Here we hypothesise that these two receptors constitute suitable targets for GRK-mediated phosphorylation (GRK2, 3, 5, and 6 in the case of V2R and GRK5 and 6 in case of AT1R) even in their inactive conformations (this has also been shown before: (Li et al., 2015) and is further supported by pharmacological stabilisation of inactive-like receptor conformations performed in the revised **Supplementary Figure 12**), given that the GRK expression levels are elevated. We do not want to make the claim that this will happen in all physiological tissues and these data should rather be seen as a characterisation of the molecular capabilities of the GPCR downregulation machinery. Since it is clear by now, that GRK expression levels vary significantly between different cell types (Matthees et al., 2021) and especially in pathophysiological conditions, we sincerely believe that this finding could aid in the explanation of pathophysiological alterations of GPCR signaling.

5. Minor, but the plot for Supp Fig 1e has out of order residue locations. The authors appear to be separating S263 from the others as a non-GRK site by putting it last, but this is confusing to anyone who looks at the figure before the text (as many do), and it would be equally separate if plotted first, as it would be if sorted by receptor residue number.

We thank the reviewer for his or her insightful comment, accurately depicting why we chose to display the data “out of amino acid order”. To address this, we changed the sequence of shown results to lead with the S363 data shown in first position, as suggested by the reviewer.

Reviewer #3 (Remarks to the Author):

In this study Drube and co-authors have developed a platform to investigate the role of individual GRKs in GPCR-arrestin interactions. They have used a systematic CRISPR/Cas9 approach to generate an impressive collection of cell lines that included single, double, triple and quadruple KO of the most ubiquitously expressed GRKs. They have then combined the use of these cells with a BRET system to assess GPCR-arrestin interactions to investigate i) the role of endogenous GRKs in promoting such interactions and ii) the effect of individual GRK2 overexpression in GPCR regulation. In doing these experiments on a group of twelve GPCRs they have encountered several interesting findings, and, perhaps most importantly, they have shed light into the determinants (or lack of thereof) guiding receptor phosphorylation and its subsequent intracellular consequences. This study has the potential to extend beyond the “phosphorylation barcode” hypothesis for GPCRs.

Although the manuscript has a huge amount of data, the authors make an effort to present it clearly and the balance of data between main figures and supplementary is adequate.

We kindly thank the reviewer for his or her comprehensive summary of our manuscript. We are grateful for the overall positive evaluation of our work and its implications on the “phosphorylation barcode” hypothesis.

Comments:

- 1- In line 147, the authors state that “data which could be described by a curve-fit will be further interpreted as functional recruitment.” This seems to be a weak criterium to classify responses and is particularly apparent in some panels in Supp Figs 3 and supported by the bar graphs in supp fig4. GraphPad Prism will sometimes fit curves over flat lines; thus, the CI of the individual curve fits, or statistical analysis on maximal responses compared to vehicles in the instances where the authors see a “response” should be provided to provide confidence on such observations.

*We thank the reviewer for pointing out this initial lack of our conceptual clarity. We unconditionally agree with the reviewer and admit that the solution we chose to implement in the initial submission of the manuscript was a shortcut and not appropriately supporting our biological interpretations. To address these shortcomings, we provided tables that contain the statistical analysis of BRET changes that were measured at saturating ligand concentrations compared with the vehicle-stimulated condition, for all performed β -arrestin recruitment assays. These data can be accessed via **Supplementary Table 1**. Furthermore, we eliminated curves that were fitted for concentration-dependent β -arrestin association events, which did not show a significant difference between vehicle and stimulated conditions. Therefore, the reader is only presented curve-fits for datasets with a confirmed statistical difference between the highest ligand concentration and vehicle addition. Additionally, we adjusted our initial phrasing in line 147 (**now lines 157-163**) to accurately portray this more elaborate way of interpreting NanoBRET recruitment data.*

- 2- In line 190 the authors discuss the relative efficacy of the endogenous ligands norepinephrine and epinephrine. The data in supp fig 2 seems to show that these two agonists are partial relative to Iso. This is in contrast to reference 27, where epinephrine is a full agonist. It is noted that the experimental set up in ref 27 and in the current manuscript is very different and perhaps the source of this discrepancy? In line with this, the authors should comment on the choice of assay to monitor GPCR-arrestin interaction there are multiple formats to detect such interactions (BRET, NanoBiT, FRET, DiscoveRx, Tango...) and assay format may impact data interpretation.

We are grateful for the reviewer’s precise assessment of our presented data and previous publications. Moreover, the reviewer accurately described the reason for the apparent discrepancy between the

datasets in question: the utilised experimental setups. The data shown in Reiner et al. 2010 Figure 5 B indeed indicate that epinephrine induces isoproterenol-like β -arrestin2 recruitment at the b2AR. The addressed discrepancy actually does not arise from the use of different resonance energy transfer measurement systems (NanoBRET in the current manuscript vs CFP/YFP FRET in Reiner et al. 2010), as both systems feature the ratiometric assessment of protein–protein interactions and share similar advantages and disadvantages. Yet, the measurements in Reiner et al. 2010 were not conducted by assessment of an ensemble of cells, but on a single-cell level using a specialised FRET microscope setup. Thus, measurements of individual cells feature vastly different signal amplitudes and have to be directly compared with or normalised to the reference agonist response. Hence, the displayed data in Reiner et al. 2010 Figure 5 B was recorded by an initial reference stimulation with isoproterenol (“first pulse”), followed by a wash-out and the subsequent measurement of the epinephrine / norepinephrine / etc. response (“second pulse”). The initial receptor activation with isoproterenol will lead to a robust phosphorylation of the available b2AR molecules, which will not be completely reversed by the following wash-out. Thus, the normalised recruitment data might display a minor over-interpretation for the tested endogenous agonists, as the b2AR in this experimental setup will still feature a certain degree of isoproterenol-induced phosphorylation.

Concerning the assay choice in the manuscript at hand, we profoundly believe that the NanoBRET measurement system is superior to most other systems, especially for the assessment of concentration-dependent protein–protein interactions. NanoBRET utilises optimised BRET donor (NanoLuc) and acceptor (Halo-Tag-618) molecules, which will lead to an improved sensitivity of the assay, as compared to other BRET systems. Moreover, as BRET does not require the introduction of an external light source, NanoBRET features an improved signal-to-noise ratio, as compared to FRET techniques. The NanoBiT technology seems to yield robust recruitment data as well, but the inherent affinity of the luciferase fragments might obscure the proper interpretation of recorded data. Additionally, the NanoBiT measurement system is not ratiometric. Hence, we would not have been able to monitor more intricate processes, like the described ligand-independent pre-coupling of arrestins to the e.g. V2R upon GRK overexpression. The PathHunter® assay, which is commercialised by DiscoverX, shares both disadvantages of the NanoBiT technology, as it utilises enzyme complementation as well. Finally, the mentioned Tango assay relies upon genetic amplification of the read-out signal via a cleaved transcription factor and measurement of enzyme activity after several hours. Hence, the assay does not yield any kinetic data, and even though we did not analyse association kinetics in this manuscript, monitoring of the association data over time was essential for the optimisation of the NanoBRET assay.

- 3- If I understand correctly, data plotted in the concentration response curves and bar graphs in Fig 3a and b respectively, should be related. Roughly the ratio stimulated/baseline in the bar graph should approx give the saturation asymptote of the CRC. This doesn't seem to be the case for DeltaQ-GRK +EV V2R barr1. According to the protocols the difference between these graphs is the additional step of vehicle normalisation in the curve that is not present in the bar graph. Can the authors clarify? Is the vehicle effect then very different in that specific condition?

We thank the referee for his or her accurate portrayal of our data processing and normalisation. Indeed, the presented bar charts in Figure 3a imply that addition of vehicle reduces the measured BRET ratios, explaining the relatively higher BRET change plotted in the concentration response curve. After thorough review of our raw data, we can confirm that all V2R measurements show this behaviour, albeit to different extents. Nevertheless, the accurately portrayed Δ net BRET change shows a robust AVP concentration-dependent increase in BRET, which, we would like to argue, convincingly depicts the agonist-induced recruitment of β -arrestins to the V2R. Additionally, all plotted data have been deposited with the Nature Communications Data Availability services.

Of note, we are aware that showing both depictions of measured BRET ratios (concentration response and baseline/stimulated) is not an ideal solution to clarify the observed biological findings, but rather necessary, since the V2R and AT1R show this rare increase in baseline BRET upon overexpression of specific GRK isoforms.

- 4- The mechanism proposed by the authors whereby individual kinases facilitate “constitutive” engagement of arrestins to V2R and AT1Rs is very interesting. How do the authors hypothesise the kinases mediate receptor phosphorylation that regulates arrestin recruitment? Can it be constitutive activity of the receptor? Perhaps this would explain the b2V2 chimera result, whereby no constitutive activity is present from b2 as opposed from V2 and hence no basal phosphorylation? Does V2R show constitutive activity? If so, it would be interesting to test whether an inverse agonist prevents the raise in baseline observed upon overexpression of individual GRKs. Alternatively, or in parallel, use of kinase dead GRKs for V2R as with the AT1R would shed some insight into this question.

*We thank the reviewer for his or her shared interest and the insightful comments and suggestions concerning ligand-independent interactions between β -arrestins and the V2R or AT1R. We agree that explanation of this intriguing mechanism could still use the support by some additional investigation. Hence, we performed both sets of experiments (GRK kinase-dead and the inverse agonist/antagonist approach) for both receptors. With the use of the appropriate kinase-dead GRK mutants (V2R: **Supplementary Figure 8**; AT1R: **Supplementary Figure 9**, note the AT1R data was already present in the initial submission), we found that they were not able to change the measured baseline or ligand-stimulated BRET ratios in comparison to Δ Q-GRK + EV. These experiments show that the found “pre-coupling” mechanism is essentially mediated by the kinase activity of overexpressed GRKs. Since we additionally show these constitutively engaged complexes are mostly persistent in a “hanging” configuration, our experiments suggest that these interactions are, in fact, stabilised by C-terminal GPCR phosphorylation. As the reviewer aptly points out, this ligand-independent phosphorylation might be evoked by constitutive activity of said receptors. Hence, we used Tolvaptan and Losartan to stabilise the inactive conformations of V2R and AT1R (**Supplementary Figure 12**), respectively, and prevent individual receptor molecules to sample active-like conformations. The overnight incubation with Losartan abolished the AngII-induced β -arrestin2 recruitment to the AT1R in all tested conditions. Still, the significantly elevated baseline BRET ratio (as compared to Δ Q-GRK + EV), induced by the overexpression of GRK6, remained unchanged. Here, we conclude that the inactive conformation of the AT1R represents a suitable target for phosphorylation by overexpressed GRK6. Interestingly, Tolvaptan treatment of V2R expressing cells reduced the measured baseline BRET ratios for GRK2/6 overexpression, yet they are still significantly increased as compared to Δ Q-GRK + EV. This apparent reduction in GRK-mediated pre-coupling of β -arrestin2 could suggest that conformational flexibility of the V2R plays a role in its ligand-independent phosphorylation. However, analogous to the AT1R, also the Tolvaptan-stabilised conformation of the V2R seems to be phosphorylated by overexpressed GRKs, which translates into a constitutive engagement of β -arrestin2.*

- 5- Line 321 - “We observed the most prominent difference between angiotensin II (AngII)-induced β -arrestin1 and 2 recruitment to the AT1R, in Δ Q-GRK, as this condition features a pronounced higher recruitment of β -arrestin2 (Supplementary Figure 3g)” – this statement is better supported by the data in Supp Fig 4.

*We agree with the referee and added a cross reference to **Supplementary Figure 4** to support the mentioned statement (**line 341**). Still, we chose to cross reference **Supplementary Figure 3g**, since this is the only representation that displays the accurate fold change in BRET signal corrected for differences in measured baselines and vehicle addition.*

- 6- The paragraph describing GRK2/3 and PKC effects on AT1R-arrestin interactions (starting line 350) needs to make it clearer that is its specifically referring to ligand-induced interactions. While the authors are very familiar to the terminology they use, the reader may need a bit more guidance.

We thank the referee for pointing out the ambiguity of our initial phrasing. We agree that the differential and GRK-specific effects that we describe might be hard to follow, especially since we report findings

that concern ligand-activated and ligand-free settings. Since, the submitted results section was presented in such a concise way in an attempt to keep the manuscript focussed and streamlined, we gladly expanded our explanations and changed the paragraph according to the reviewer's suggestion (lines 362-374).

- 7- Line 414 "While the V2R and AT1R are both regulated by all tested GRKs, they exhibited substantial β -arrestin pre-coupling and therefore constitute a subgroup of GRK2/3/5/6-regulated receptors". This statement should clarify that the pre-coupling was observed upon GRK overexpression.

We thank the reviewer for indicating the inaccuracy in this section. This is indeed a very important point and in agreement with the reviewer's suggestion, we clarified the wording (line 433) and double-checked the entire manuscript for other sections that mention "pre-coupling" of β -arrestins to add this crucial information.

- 8- A considerable section of the discussion attempts to find correlations between phosphorylation motifs and GRK specificity. A recent attempt to identify different functional classes of phosphorylation sites suggested the existence of "key", "inhibitory" and "modulatory" phosphorylation sites for arrestin binding (Mayer et al Nat Comms 2019). Similarly, computational and biophysical studies on GPCR phosphorylation patterns have recently addressed the question of "phosphorylation barcodes" (e.g. Latorraca et al., Cell 2020). Some discussion around the context of this previous work and the current results should be included.

We thank the reviewer for this suggestion and bringing the two mentioned publications to our attention. Both of them investigate how arrestin conformational changes are modulated by specific C-terminal GPCR phosphorylation patterns. Yet, the definitive influence of the seven-transmembrane core and the occurrence of GRK-specific phosphorylation patterns have been mostly neglected. Hence, we believe that Drube et al. represents a complimentary study, which shows that arrestin functions cannot be comprehensively assessed with the use of phosphopeptides only.

During the course of data collection for this manuscript, we were especially intrigued by the alignment presented in Figure 8 of Mayer et al.. The abolished concentration-dependent β -arrestin recruitment to the GRK5/6-phosphorylated AT1R was an initially puzzling result, which we were only able to appropriately interpret as "pre-coupling" after analysis of the raw baseline and stimulated BRET ratios. Interestingly, the AT1R is listed in the mentioned alignment and thus we proceeded by creating alanine mutants of the proposed "inhibitory" phosphorylation sites. The GRK-specific recruitment data for these three mutant AT1R constructs (S347A, S348A, and the double mutant S347A/S348A) are now depicted in **Supplementary Figure 14**. Neither of the three mutant receptors showed an altered "pre-coupling" behavior upon GRK5 or 6 overexpression, yet mutation of the S347 phosphorylation site reduced GRK2 and 3-specific β -arrestin2 recruitment to the level measured in Δ Q-GRK. Thus, we can exclude that these two phosphorylation sites play an inhibitory role – especially for the recruitment of β -arrestin2. We are aware that this set of experiments is far from comprehensive, yet they aid our argument, showing that the cellular functionality of arrestins cannot necessarily be delineated from experiments conducted with synthesized phosphopeptides.

Both suggested publications are now mentioned in the **discussion** section (lines 545-573) and reviewed in the light of our novel findings.

- 9- Minor comment: Line 494 – "whereas" should be "whereby"?

We thank the reviewer for providing this editorial correction. The sentence has been changed according to the reviewer's suggestion.

References:

- Li, L., Homan, K. T., Vishnivetskiy, S. A., Manglik, A., Tesmer, J. J., Gurevich, V. V., & Gurevich, E. V. (2015). G Protein-coupled Receptor Kinases of the GRK4 Protein Subfamily Phosphorylate Inactive G Protein-coupled Receptors (GPCRs). *J Biol Chem*, 290(17), 10775-10790. <https://doi.org/10.1074/jbc.M115.644773>
- Matthees, E. S. F., Haider, R. S., Hoffmann, C., & Drube, J. (2021). Differential Regulation of GPCRs-Are GRK Expression Levels the Key? *Front Cell Dev Biol*, 9, 687489. <https://doi.org/10.3389/fcell.2021.687489>
- Reichel, M., Weitzel, V., Klement, L., Hoffmann, C., & Drube, J. (2021). Suitability of GRK antibodies for individual detection and quantification of GRK isoforms in western blots. *bioRxiv*, 2021.2010.2026.465910. <https://doi.org/10.1101/2021.10.26.465910>

REVIEWER COMMENTS

Reviewer #1 (Remarks to the Author):

The authors created a panel of HEK293 cells lacking individual GRKs and their various combinations, including cells lacking GRK2/3/5/6. All these cell lines would be very useful for the GPCR field. The authors used these cells +/- overexpression of individual GRKs and PKC inhibition to determine relative importance of individual GRK subtypes in b-arrestin1/2 binding to a wide panel of GPCRs belonging to classes A and B that couple to different G proteins. In the revised manuscript the authors provide nuanced interpretation of the results of these extensive meticulously performed studies that reveal complexities of GPCR regulation by GRKs and b-arrestins. As presented now, the data are of great interest to the GPCR field. A few editorial revisions would further improve the manuscript.

In particular:

line 70 “downregulation of photoreceptors” should be “shutoff of photopigment signaling”;

line 71, GRK4 is also highly expressed in the heart;

line 73, “photoreceptor desensitisation” should be “photopigment desensitization”;

line 74, eliminate “respectively” (in cones there is a lot more arrestin-1 than arrestin-4);

line 77, “accommodate for” “should be “explain” or something similar;

line 88, multi-step arrestin-GPCR binding model was proposed back in 1993 (J Biol Chem. 1993 Jun 5;268(16):11628-38); in fact, the authors data suggest that the hypothesis proposed in 1993 that receptor phosphorylation and activation are independent equal triggers of arrestin binding, so that either can be engaged by the arrestin first;

line 92, “on receptor phosphorylation” should be “in receptor phosphorylation”;

line 124 and below “ μ -124 opioid receptor” is usually abbreviated MOR, not MOP;

to conform to American English, throughout the paper, “internalisation” should be “internalization”.

“utilising” should be “utilizing”. “localisation” should be “localization”, etc.;

for consistency, throughout the paper “Western blot” should be either capitalized or not;

not all receptors used in this study have GRK phosphorylation sites in their C-terminus: e.g., M2R has none in the C-terminus, as all GRK sites are localized to the 3rd cytoplasmic loop, so the authors should modify the text accordingly;

line 387, delete “on”;

line 1163, “occupation” should be “occupancy”.

Reviewer #2 (Remarks to the Author):

The authors have extensively revised their manuscript. Specifically, they have added new experiments in response to reviewer requests and clarified statistics throughout. These changes address the concerns raised during review.

There remain some minor language/clarity issues in the revised text:

Line 68, “comprises” is incorrect, and “contains” is a better word choice.

Line 160, “only those data sets, which show” should read “only those data sets that show”

Line 583, “besides C-terminal or IL3 receptor phosphorylation” would be clearer if stated as “besides receptor phosphorylation at the C-terminal or IL3”

Reviewer #3 (Remarks to the Author):

The authors have thoroughly addressed my comments. They have clarified ambiguities and adjusted interpretations with their new data. I suggest acceptance of the manuscript.

2nd round of revision – point-to-point reply to the reviewers’ comments:

We thank the three referees as well as the editor for their positive comments and general appreciation of our study. In the final revision of our manuscript we addressed all points that were raised by the reviewers and included all editorial requests.

Reviewer #1 (Remarks to the Author):

The authors created a panel of HEK293 cells lacking individual GRKs and their various combinations, including cells lacking GRK2/3/5/6. All these cell lines would be very useful for the GPCR field. The authors used these cells +/- overexpression of individual GRKs and PKC inhibition to determine relative importance of individual GRK subtypes in b-arrestin1/2 binding to a wide panel of GPCRs belonging to classes A and B that couple to different G proteins. In the revised manuscript the authors provide nuanced interpretation of the results of these extensive meticulously performed studies that reveal complexities of GPCR regulation by GRKs and b-arrestins. As presented now, the data are of great interest to the GPCR field. A few editorial revisions would further improve the manuscript.

We thank the referee for his or her comprehensive summary of our study. We especially appreciate the overall positive assessment of our revised manuscript and its impact on GPCR research.

In particular

1. line 70 “downregulation of photoreceptors” should be “shutoff of photopigment signaling”;

We acknowledge the reviewer’s suggestion and the text has been changes as proposed.

2. line 71, GRK4 is also highly expressed in the heart;

This statement is now included in the final version of the manuscript.

3. line 73, “photoreceptor desensitisation” should be “photopigment desensitization”;

To be in line with the 1st suggestion of the reviewer, this sentence has also been changed.

4. line 74, eliminate “respectively” (in cones there is a lot more arrestin-1 than arrestin-4);

We thank the referee for making us aware of this interesting point. The word “respectively” has been removed from the sentence.

5. line 77, “accommodate for” “should be “explain” or something similar;

This statement has been changed to “to explain this apparent imbalance”, according to the reviewer’s suggestion.

6. line 88, multi-step arrestin-GPCR binding model was proposed back in 1993 (J Biol Chem. 1993 Jun 5;268(16):11628-38); in fact, the authors data suggest that the hypothesis proposed in 1993

that receptor phosphorylation and activation are independent equal triggers of arrestin binding, so that either can be engaged by the arrestin first;

We thank the reviewer for pointing out this interesting study. We included the reference at the suggested position in the introduction.

7. line 92, “on receptor phosphorylation” should be “in receptor phosphorylation”;

This statement has been changed, according to the referee’s suggestion.

8. line 124 and below “ μ -124 opioid receptor” is usually abbreviated MOR, not MOP;

In accordance with IUPHAR guidelines, we politely request to keep the abbreviation for the μ opioid receptor unchanged. We would like to continually refer to this receptor as MOP, unless the Journal or editor insists on this change.

9. to conform to American English, throughout the paper, “internalisation” should be “internalization”. “utilising” should be “utilizing”. “localisation” should be “localization”, etc.;

In accordance with Nature Communications, we prepared the entire manuscript using British English. Hence, we respectfully request not to perform the suggested changes.

10. for consistency, throughout the paper “Western blot” should be either capitalized or not;

We thank the reviewer for finding this inconsistency in the manuscript. The western blot technique is now referred to as “western blot” throughout the final version of the manuscript.

11. not all receptors used in this study have GRK phosphorylation sites in their C-terminus: e.g., M2R has none in the C-terminus, as all GRK sites are localized to the 3rd cytoplasmic loop, so the authors should modify the text accordingly;

*We thank the reviewer for this comment. After a thorough review of our manuscript, all sections of the manuscript that describe the general phosphorylation-dependency of arrestin interactions with GPCRs, now feature the phrasing “phosphorylated intracellular receptor domains” (lines: 309, 453, 496). Of note, a summary describing the exact length of said intracellular domains as well as the number of included serine and threonine residues can be found in **Supplementary Table 2**.*

12. line 387, delete “on”;

The sentence has been changed, according to the reviewer’s suggestion.

13. line 1163, “occupation” should be “occupancy”.

We thank the reviewer for pointing this out – we changed the sentence accordingly.

Reviewer #2 (Remarks to the Author):

The authors have extensively revised their manuscript. Specifically, they have added new experiments in response to reviewer requests and clarified statistics throughout. These changes address the concerns raised during review.

We thank the reviewer for her or his positive feedback regarding our revised manuscript. We agree with the referee's assessment, as the added experiments improved on our existing work.

some minor language/clarity issues in the revised text

1. Line 68, "comprises" is incorrect, and "contains" is a better word choice.

We thank the referee for this comment - the wording has been changes as suggested.

2. Line 160, "only those data sets, which show" should read "only those data sets that show"

According to the reviewer's suggestion the sentence has been changed .

3. Line 583, "besides C-terminal or IL3 receptor phosphorylation" would be clearer if stated as "besides receptor phosphorylation at the C-terminal or IL3"

We agree with the reviewer and adjusted the phrasing, as suggested.

Reviewer #3 (Remarks to the Author):

The authors have thoroughly addressed my comments. They have clarified ambiguities and adjusted interpretations with their new data. I suggest acceptance of the manuscript.

We kindly thank the reviewer for her or his positive assessment and the recommendation for acceptance.